# Discovery of first-in-class reversible dual small molecule inhibitors against G9a and DNMTs in hematological malignancies

Edurne San José-Enériz[1,*], Xabier Agirre[1,*], Obdulia Rabal[2,*], Amaia Vilas-Zornoza[1], Juan A. Sanchez-Arias[2], Estibaliz Miranda[1], Ana Ugarte[2], Sergio Roa[1], Bruno Paiva[1], Ander Estella-Hermoso de Mendoza[2], Rosa María Alvarez[2], Noelia Casares[3], Victor Segura[4], José I. Martín-Subero[5], François-Xavier Ogi[6], Pierre Soule[6], Clara M. Santiveri[7], Ramón Campos-Olivas[7], Giancarlo Castellano[5], Maite Garcia Fernandez de Barrena[3], Juan Roberto Rodriguez-Madoz[1], Maria José García-Barchino[1], Juan Jose Lasarte[3], Matias A. Avila[3], Jose Angel Martinez-Climent[1], Julen Oyarzabal[2] & Felipe Prosper[1,8]

The indisputable role of epigenetics in cancer and the fact that epigenetic alterations can be reversed have favoured development of epigenetic drugs. In this study, we design and synthesize potent novel, selective and reversible chemical probes that simultaneously inhibit the G9a and DNMTs methyltransferase activity. *In vitro* treatment of haematological neoplasia (acute myeloid leukaemia-AML, acute lymphoblastic leukaemia-ALL and diffuse large B-cell lymphoma-DLBCL) with the lead compound CM-272, inhibits cell proliferation and promotes apoptosis, inducing interferon-stimulated genes and immunogenic cell death. CM-272 significantly prolongs survival of AML, ALL and DLBCL xenogeneic models. Our results represent the discovery of first-in-class dual inhibitors of G9a/DNMTs and establish this chemical series as a promising therapeutic tool for unmet needs in haematological tumours.

[1] Area de Hemato-Oncología, Centro de Investigación Médica Aplicada, IDISNA, Ciberonc, Universidad de Navarra, Avenida Pío XII, 55 31008 Pamplona, Spain. [2] Small Molecule Discovery Platform, Molecular Therapeutics Program, Center for Applied Medical Research, University of Navarra, Avenida Pío XII, 55 31008 Pamplona, Spain. [3] Area de Terapia Génica y Hepatología, Centro de Investigación Médica Aplicada, Universidad de Navarra, Avenida Pío XII, 55 31008 Pamplona, Spain. [4] Unidad de Bioinformática, Centro de Investigación Médica Aplicada, Universidad de Navarra, Avenida Pío XII, 55 31008 Pamplona, Spain. [5] Departamento de Fundamentos Clínicos, Universitat de Barcelona, Institut d'Investigacions Biomèdiques August Pi i Sunyer, Centre Esther Koplowitz, C/ Rosello 153 2nd floor 08036 Barcelona, Spain. [6] Nanotemper Technologies GmbH, Flößergasse 4, Munich, Germany. [7] Spectroscopy and NMR Unit, Spanish National Cancer Research Center (CNIO), C/ Melchor Fernández Almagro, 3 28029 Madrid, Spain. [8] Departamento de Hematología, Clínica Universidad de Navarra, Universidad de Navarra, Avenida Pío XII, 36 31008 Pamplona, Spain. * These authors contributed equally to this work. Correspondence and requests for materials should be addressed to J.O. (email: julenoyarzabal@unav.es) or to F.P. (email: fprosper@unav.es).

Therapy of haematological malignancies is undergoing a paradigm shift away from the traditional chemotherapy towards the targeting of proteins driving the cancer phenotype[1]. One of such novel therapeutic approaches relies on the use of epigenetic drugs, which aim to reverse critical epigenetic events underlying human cancer pathogenesis, particularly abnormalities in DNA methylation and histone modifications[2]. The recent approval of two types of epigenetic drugs for poor-prognosis haematological tumours, namely the DNA methyltransferases inhibitors azacitidine and decitabine and the histone deacetylase inhibitors (HDACi) vorinostat and panobinostat have shown promising clinical benefits for patients who are ineligible or refractory to current therapies[3,4]. In addition, novel epigenetic drugs targeting histone lysine methylation are being developed[5], including inhibitors of the histone methyltransferase G9a, also known as EHMT2. G9a is overexpressed in many tumours, and the methylation of its target lysine 9 of histone 3 (H3K9) is associated with transcriptional silencing[6–8]. Several studies have shown that inhibition of G9a expression decreases cancer cell proliferation[9], delays disease progression[10] and blocks tumour metastasis[6,7].

Interestingly, G9a physically interacts with DNA methyltransferase-1 (DNMT1) to coordinate DNA and histone methylation during cell division[11] promoting transcriptional silencing of target genes[12]. In this sense, reduction of both DNA and H3K9 methylation levels leads to reactivation of tumour suppressor genes and inhibits cancer cell proliferation[13,14]. Therefore, we postulated that small molecule inhibitors simultaneously targeting the methyltransferase activity of G9a and DNMTs might represent an improved approach in cancer therapeutics.

Here, we design potent novel, selective and reversible dual small molecules against G9a and DNMTs activity. The lead compound CM-272 inhibits cell proliferation and promotes apoptosis in different haematological neoplasias (AML, ALL and DLBCL), inducing interferon-stimulated genes and immunogenic cell death. CM-272 also prolongs survival of AML, ALL and DLBCL xenogeneic models. These compounds represent a novel and promising approach for treating a broad series of human tumours with poor prognosis.

## Results

**Novel substrate-competitive dual inhibitors of G9a and DNMTs.** First, to show that simultaneous inhibition of G9a and DNMTs could improve the treatment of cancer, we treated the OCI-AML-2 AML cell line with an inhibitor for G9a (A-366) and another inhibitor for DNMTs (decitabine). This analysis demonstrated that the combination of these two inhibitors presented a synergism in reducing the growth of leukaemic cells (Supplementary Fig. 1a,b). We obtained similar results using specific siRNAs against G9a and DNMT1, showing that the combination of G9a and DNMT1 siRNAs induced a significantly greater inhibition of cell proliferation in comparison with any siRNA separately (Supplementary Fig. 1c). These results indicate the synergistic effect of simultaneous inhibition of methyltransferase activity of G9a and DNMTs on proliferation of leukaemic cells.

Knowledge- and structure-based approaches guided us to design first-in-class dual inhibitors of G9a and DNMTs methyltransferase activity. On the basis of reported structure–activity relationships (SAR) data from G9a substrate-competitive inhibition[15–17] together with the available structural information, X-ray co-crystal structures of a G9a-UNC0638 complex (PDB 3RJW)[18], and of a DNMT1-hemimethylated CpG DNA complex (PDB 4DA4)[19], we designed and synthesized compounds to interact both with G9a and DNMT1. We directed our efforts towards the identification of a ligand–

receptor interaction at substrate-binding sites, that is, histone 3 (H3) and DNA competitive. Consequently, a novel chemical series of 4-aminoquinolines, bearing key chemical functionalities that cover critical pharmacophoric features, were designed (Fig. 1 and Supplementary Fig. 2a,b). Detailed exploration of this series, which included more than 100 compounds, showed their dual activity against G9a and DNMTs (information regarding this novel series is reported in our patent[20]). To validate our design strategy, 4-oxyquinolines that lose a key hydrogen bond interaction with Asp1083 G9a (secondary amine was replaced by ether) were synthesized. The corresponding primary activity of one of such compounds, CM-1021, was reduced in more than 2.5 log units (Fig. 1a and Supplementary Fig. 2c) and therefore, we did not progress further with this particular chemical series, validating our synthesis strategy. For primary screening assays, a previously described time-resolved fluorescence resonance energy transfer (TR-FRET) technology was used to measure biochemical inhibition of G9a enzymatic activity[21]. In addition, we implemented a new assay also based on TR-FRET technology to analyse the DNMT1 enzymatic activity. As decision-making criteria, we focused on compounds fulfilling pre-established dual activity criteria (IC$_{50}$ versus G9a $<100$ nM and versus DNMT1 $<500$ nM): potent enough against both targets (nanomolar range). Only molecules reaching this threshold were selected for further studies.

Extensive multifactorial optimization of these molecules through medicinal chemistry, followed by ADME studies (absorption, distribution, metabolism and excretion properties), and by cardiovascular safety and toxicity analyses, pointed out to CM-272 and CM-579 as the two lead pharmacological tool compounds (Fig. 1a; Supplementary Tables 1 and 2). We observed that CM-272 and CM-579 showed IC$_{50}$ values against G9a of 8 and 16 nM, respectively, and values versus DNMT1 activity of 382 and 32 nM, respectively; confirmed by corresponding orthogonal biochemical assays, $^3$H-SAM (tritiated SAM) (Fig. 1a, Supplementary Fig. 2c and Supplementary Table 3) as well as by direct binding measurements using microscale thermophoresis as biophysical method ($K_d$ of CM-579 versus DNMT1 is 1.5 nM) (Supplementary Fig. 3). Therefore, our novel molecules showed similar IC$_{50}$ values for G9a and DNMT1 to those of the corresponding reference compounds against G9a (BIX-01294 and UNC0638)[15,18] and to those of the irreversible DNMT inhibitors azacitidine and decitabine[22] (Fig. 1b and Supplementary Table 3). Of note, CM-272 and CM-579 also inhibited DNMT3A (IC$_{50s}$, 85 and 92 nM, respectively) and DNMT3B (IC$_{50s}$, 1,200 and 1,000 nM, respectively) and also GLP (IC$_{50s}$, 2 nM (or 7 nM using the radioligand-binding assay) and $>10,000$ (or 67 nM using the radioligand binding assay), respectively) (Supplementary Table 4a,b). For both G9a and DNMT targets and according to the designed strategy and docking studies, plausible binding modes suggested that CM-272 and CM-579 occupy the substrate-binding groove (H3 or DNA), achieving key interactions, and do not interact with the S-adenosyl-L-methionine (SAM)-binding pocket (Fig. 1c,d and Supplementary Fig. 2a,b), despite being both proteins structurally divergent at the substrate-binding site (Supplementary Fig. 4). Indeed, experimental competition studies confirmed that CM-272 and CM-579 are substrate-competitive inhibitors of G9a and DNMT, displaced their corresponding substrates but not the cofactor SAM (Fig. 1e–h; Supplementary Fig. 2d–g and Supplementary Tables 5–8). We additionally assessed the inhibitory activity of CM-272 and CM-579 over a wide range of 37 enzymes implicated in the regulation of epigenetic mechanisms. This analysis underscored a high selectivity of CM-272 and CM-579 against their specific targets, suggesting their minimal promiscuity against other SAM-dependent

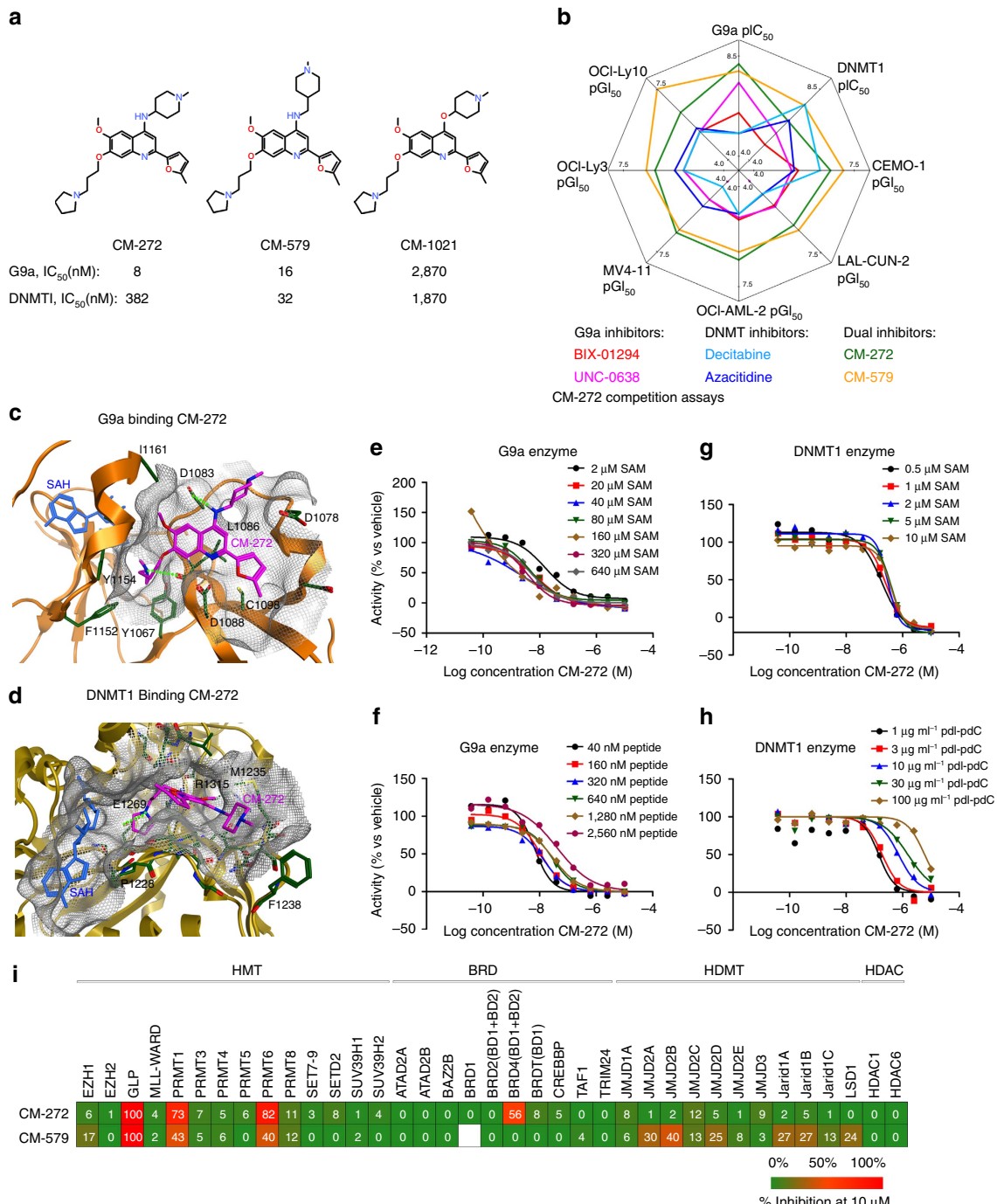

**Figure 1 | Small molecules with a dual inhibitory activity against G9a and DNMT.** (**a**) Chemical structure of CM-272, CM-579 and CM-1021 together with their $IC_{50}$ values against G9a and DNMT1 activity. (**b**) Radar plot showing the *in vitro* G9a and DNMT1 inhibitory potencies (expressed as $pIC_{50}$) and the growth inhibitory potencies (expressed as $pGI_{50}$) against different cell lines (AML: MV4-11 and OCI-AML-2: ALL: CEMO-1 and LAL-CUN-2 and DLBCL: OCI-Ly10 and OCI-Ly3) for compounds CM-272, CM-579, selective G9a inhibitors (BIX-01294 and UNC-0638) and irreversible DNMT inhibitors (azacitidine and decitabine). (**c,d**) Plausible binding mode of CM-272 (in pink, blue and red sticks) into the histone binding groove of G9a (**c**) and the DNA binding groove of DNMT1 (**d**). CM-272 does not interact with the corresponding SAM binding pockets (SAH shown in blue sticks). For G9a, the 7-(3-pyrrolidin-1-yl) propoxy side chain of CM-272 interacts with the lysine binding channel. In the case of DNMT1, this side chain is predicted to overlay with the DNA cytosine, occupying the catalytic pocket and interacting with the catalytic glutamate E1269 (mouse DNMT1). (**e–h**) Enzymatic competition assays of CM-272: G9a competition assay with the SAM cofactor (**e**) and with the histone peptide (PepMe1) (**f**) and DNMT1 competition assay with the SAM cofactor (**g**) and with the DNA substrate (**h**). (**i**) Selectivity profile of CM-272 and CM-579 at 10 μM against 37 epigenetic enzyme targets from different families.

epigenetic enzymes (Fig. 1i and Supplementary Table 4). The existence of a unique helix (αZ) in the SET domain of G9a/GLP can explain the potential selectivity of these compounds over other HKMTs. In summary, we have designed and synthesized novel, selective and potent substrate-competitive dual inhibitors against methyltransferase activity of G9a and DNMTs.

**G9a/DNMTs inhibitors show potent *in vitro* cellular activity**. To determine the anti-tumour activity of simultaneous inhibition of G9a and DNMTs methyltransferase activity, we evaluated, by MTS (3-4(,5-dimethylthiazol-2-yl)-5-(3-carboxymethoxyphenyl)-2-(4-sulfophenyl)-2H-tetrazolium) assay, the effect of CM-272 and CM-579 treatment in a collection of 75 cell lines derived from human patients with a wide range of cancers. Both molecules exhibited a significant activity at nM concentration in various haematological neoplasms and also in solid tumours with limited therapeutic options such as bladder and pancreatic cancer (Supplementary Data 1). To establish, as proof of concept, the potential therapeutic value of CM-272 and CM-579 we focused on haematologic malignancies which are commonly refractory to currently available chemo-immunotherapeutic strategies, namely acute myeloid leukaemia (AML), adult acute lymphoblastic leukaemia (ALL) and diffuse large lymphoma (DLBCL) of activated B-cell type. This decision was further based on previous studies from our group showing: (1) an association between repressive histone modifications and DNA methylation with poor prognosis in ALL, AML and DLBCL[23–26]; (2) the detection of G9a protein overexpression in leukaemia cell lines and primary samples (Supplementary Fig. 5a); (3) the decrease of cell proliferation in leukaemia cell lines upon G9a inhibition using siRNAs (Supplementary Fig. 5b,d); (4) our experience with *in vitro* and *in vivo* models of such haematological malignancies[27,28] and (5) the clinical experience with the irreversible DNMT inhibitors in AML patients[29].

We detected that the $GI_{50}$ for CM-272 and CM-579 after 48 h of treatment in ALL, AML and DLBCL-derived cell lines was in the nM range (Fig. 2a, Supplementary Fig. 6a,b and Supplementary Table 3) and was associated with a decrease in global levels of H3K9me2 and 5mC (Fig. 2b–d and Supplementary Fig. 6c), but not of other histone marks such as an H3K27me3, H3K36me3, H3K4me3, H3K79me3 and H3 acetylation (Supplementary Fig. 6d). We also observed the loss of DNA methylation in the promoter region of specific tumour suppressor genes after CM-272 treatment (Supplementary Fig. 7a,b). These results suggest that CM-272 acts *in vitro* by selective inhibition of both G9a and DNMT methyltransferase activities. To study in further detail, the biological *in vitro* effects of CM-272, we exposed the tumour cell lines to different concentrations of the drug during 1 to 3 days. We detected that CM-272 inhibited cell proliferation (Fig. 2e and Supplementary Fig. 8a), blocked cell cycle progression (Fig. 2f and Supplementary Fig. 8b) and induced apoptosis in ALL, AML and DLBCL cell lines in a dose-dependent manner (Fig. 2g and Supplementary Fig. 8c). Next, we compared the *in vitro* effects of our two dual G9a and DNMTs inhibitors with currently available single inhibitors of G9a (BIX-01294 and UNC0638) and of DNMTs (azacitidine and decitabine). Interestingly, we observed that these reference compounds were less effective than our inhibitors CM-272 or CM-579 in ALL, AML and DLBCL cells, exhibiting $GI_{50}$ values in the μM range (Fig. 1b and Supplementary Table 3). These results indicate that our dual inhibitors CM-272 and CM-579 induced a potent therapeutic response *in vitro* due to their inhibitory effect against methyltransferase activity of G9a and DNMTs.

**CM-272 induces IFN response and immunogenic cell death**. To gain additional insight into the molecular mechanisms underlying the therapeutic effect of the dual inhibitors against G9a and DNMTs, we compared the transcriptome of CEMO-1 ALL-derived cells by RNA sequencing before and after treatment with CM-272 at $GI_{50}$ concentrations. A decrease in H3K9me2 and 5mC levels was confirmed upon CM-272 exposure. Ingenuity

pathway analysis showed that cell proliferation, cell death, DNA repair and cell metabolism including TP53 and MYC-related pathways were deregulated. Interestingly, bioinformatic studies using gene set enrichment analysis (GSEA) of the RNA-seq data with false discovery rate <1%, revealed 126 differentially expressed gene probe sets after CM-272 treatment (Supplementary Data 2). We observed that 11 of these gene-sets were related to interferon (IFN) responses; in fact, the IFN response groups showed the highest number of differentially expressed gene-sets. In addition, because H3K9me2 has been reported to be an epigenetic signature of the IFN response[30], we focused on type I IFN response as a potential mechanism of CM-272 anti-tumour effects. We detected using a GSEA of the transcriptional data, based on patterns of IFN-stimulated genes (ISGs) recently published[31,32] that genes upregulated in ALL cells after CM-272 treatment were significantly enriched among the type I IFN gene set (Fig. 3a). In addition, the majority of these ISGs showed a markedly increased expression in ALL cells after CM-272 exposure (Fig. 3b). These results suggested that CM-272 induces type I IFN pathway in ALL through inhibition of G9a and DNMTs methyltransferase activity. Importantly, we observed similar results in both AML and DLBCL cells (Fig. 3a,b), pointing to a common therapeutic mechanism induced by CM-272 in different haematological tumours. Validation analysis using quantitative RT-PCR (qRT-PCR) confirmed increased expression of selected upregulated ISGs after CM-272 exposure (Fig. 3c), beginning early after treatment (12 h) and showing the highest changes in their expression at 24 and 48 h after CM-272 treatment (Supplementary Fig. 9a). H3K9me2 qChIP analyses showed that the increased expression of six of the ISGs was associated with a decrease in H3K9me2 levels in their promoters (Fig. 3d). We did not detect the H3K9me2 changes in genes without expression modulation after CM-272 treatment (Supplementary Fig. 9b,c), indicating that the epigenetic modulation after CM-272 treatment occurs in specific genomic regions and genes in leukaemic cells. Finally, we detected an increased expression of calreticulin and secretion of HMGB1 (ref. 33), two markers related to immunogenic cell death, in ALL, AML and DLBCL cells 48 h after CM-272 treatment (Fig. 3e,f). Altogether, these results suggest that the therapeutic activity of CM-272 relies on the early activation of the type I IFN response in tumour cells, potentially leading to the induction of cell autonomous immunogenic death in tumour cells.

**CM-272 exerts a potent *in vivo* activity**. Before evaluating the *in vivo* efficacy of our dual inhibitors, we examined the therapeutic window achieved by these two molecules and their pharmacokinetic (PK) parameters. Thus, we studied toxicity of CM-272 and CM-579 using the non-tumoural hepatic cell line THLE-2 ($LC_{50s}$ were 1.78 and 1.30 μM) as well as peripheral blood mononuclear cells (PBMCs) obtained from healthy donors ($LC_{50s}$ were 1.90 and 7.39 μM) (Supplementary Table 2) and compared with the *in vitro* activity against tumour cell lines (Supplementary Table 3). We detected that CM-272 and CM-579 showed an acceptable therapeutic window (around 1 log units). Before performing PK studies, we identified the maximum tolerated dose (MTD) for CM-272, which it was at $2.5\,mg\,kg^{-1}$ (intravenous, i.v.); however, for CM-579 we could not administer a dose higher than $1\,mg\,kg^{-1}$ (i.v.). PK studies that we performed using the MTD dose (Supplementary Tables 9–11), showed clearance levels for CM-579 ($5.71\,h^{-1}\,kg^{-1}$) and CM-272 ($0.91\,l\,h^{-1}\,kg^{-1}$). On the basis of these results, CM-272 was chosen for further *in vivo* studies. I.v. administration of CM-272 at $2.5\,mg\,kg^{-1}$ was the optimal dose to reach a sustained plasmatic concentration, which is above the $GI_{50}$ for CEMO-1 cells

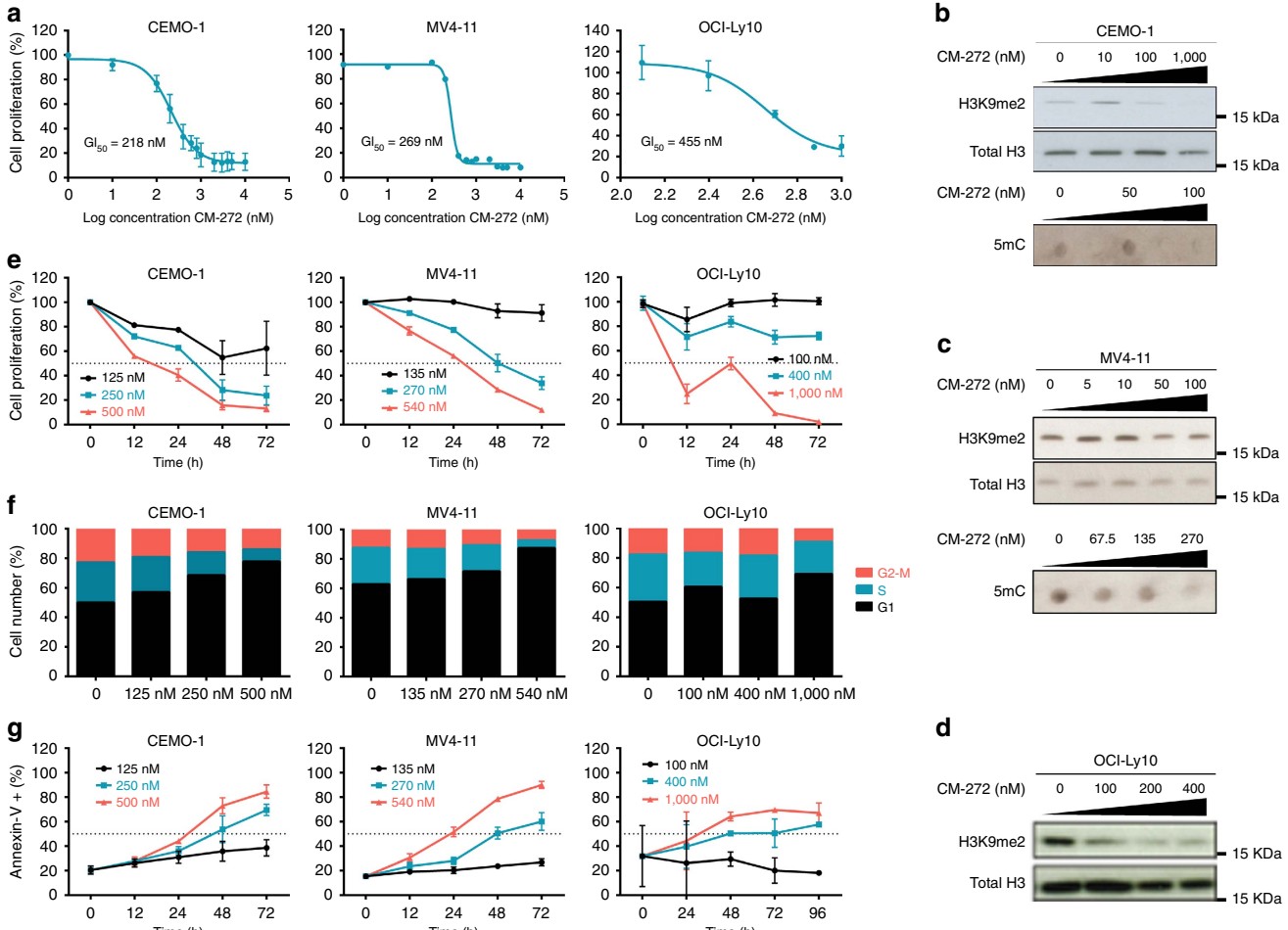

**Figure 2 | CM-272 inhibits cell proliferation and induces apoptosis.** (**a**) $GI_{50}$ values of CM-272 for CEMO-1 ALL cell line, MV4-11 AML cell line and OCI-Ly10 DLBCL cell line. (**b**) H3K9me2 and 5mC levels after CM-272 treatment with different doses for 48 h in CEMO-1 cell line. H3 total was used as loading control. (**c**) H3K9me2 and 5mC levels after CM-272 treatment with different doses for 48 h in MV4-11 cell line. H3 total was used as loading control. (**d**) H3K9me2 levels after CM-272 treatment with different doses for 48 h in OCI-Ly10 cell line. H3 total was used as loading control. (**e**) Cell proliferation time course in CEMO-1, MV4-11 and OCI-Ly10 cell lines treated with three different concentrations of CM-272 ($GI_{25}$, $GI_{50}$ and $GI_{75}$) for 12, 24, 48 and 72 h. (**f**) Cell cycle in CEMO-1, MV4-11 and OCI-Ly10 cell lines treated with three different concentrations of CM-272 ($GI_{25}$, $GI_{50}$ and $GI_{75}$) for 24 h (a representative example of three different experiments is shown). (**g**) Apoptosis time course in CEMO-1, MV4-11 and OCI-Ly10 cell lines treated with three different concentrations of CM-272 ($GI_{25}$, $GI_{50}$ and $GI_{75}$) for 12, 24, 48 and 72 h. Error bars indicate s.d. from three replicates.

and below the toxic level defined in THLE-2 cell line (Supplementary Fig. 10a). Next, we examined the potential toxicity of CM-272 in $Rag2^{-/-}\gamma c^{-/-}$ mice. Daily i.v. administration of $2.5\,mg\,kg^{-1}$ during 4 weeks, followed by a 7 days washout period, was not associated with weight loss (Supplementary Fig. 10b), other physical indicators of sickness, or major changes in haematological parameters (Supplementary Fig. 10c). In addition, histological examination of liver tissues and liver parameters did not show abnormalities in mice treated with CM-272 in comparison with control mice (Supplementary Fig. 10d,e). Overall, these results demonstrate that CM-272 is a safe molecule for administration to mice.

Next, to translate the *in vitro* anti-tumor efficacy of CM-272 to an *in vivo* model, we analysed its therapeutic effect in different xenogeneic models of haematological tumours. ALL-derived CEMO cells ($10 \times 10^6$) were injected i.v. in immunodeficient $Rag2^{-/-}\gamma c^{-/-}$ mice, which were treated with $2.5\,mg\,kg^{-1}$ of CM-272 administered daily, starting 3 days after injection and continued during 28 days. Control animals received saline solution under the same protocol. CM-272 therapy induced a statistically significant increase in overall survival (OS) in mice in comparison with control animals (median OS; $92 \pm 5.7$ days

versus $55 \pm 10.5$ days; $P = 0.0009$) (Fig. 4a). Global H3K9me2 and 5mC levels were measured in the extract from total liver. Tumour infiltration was analysed in liver homogenates by flow cytometry (FACS) analysis and showed an infiltration of 60–80% of human cells (hCD45 + ). Both marks were reduced in leukaemic cells obtained from animals after 1 week of treatment (Supplementary Fig. 10f). No significant weight loss was observed in treated animals (Supplementary Fig. 10g). We obtained similar results in a second *in vivo* replicate with CEMO-1 cells (Supplementary Fig. 11a). To analyse the dose-dependent efficacy of CM-272 *in vivo*, we repeated the same study administering $1\,mg\,kg^{-1}$ of CM-272 (Supplementary Fig. 12). We did not observe differences in the body weight of the animals (Supplementary Fig. 12a) nor significant changes in haematological parameters (Supplementary Fig. 12b) between mice treated with $1\,mg\,kg^{-1}$ or $2.5\,mg\,kg^{-1}$ of CM-272 and the control group. As expected, CM-272 plasma concentration was greater in the mice group treated with $2.5\,mg\,kg^{-1}$ of CM-272 (Supplementary Fig. 12c). However, treatment with $1\,mg\,kg^{-1}$ of CM-272 was not able to prolong survival of the mice unlike what was observed when the $2.5\,mg\,kg^{-1}$ of CM-272 was used (Supplementary Fig. 12d). These results demonstrate

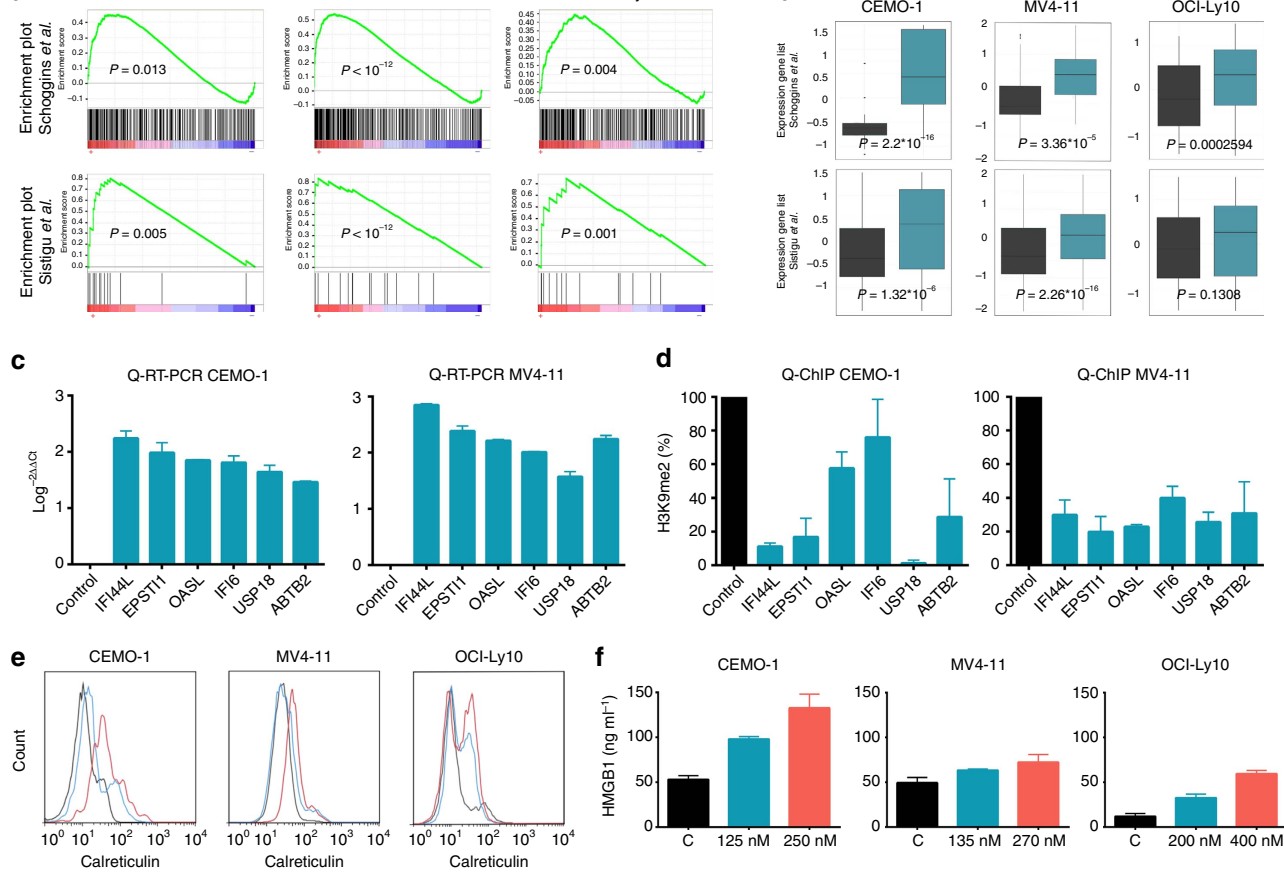

**Figure 3 | CM-272 induces type I IFN response and immunogenic cell death.** (**a**) Enrichment of ISGs published by Schoggins and colleagues[31] and Sistigu and colleagues[32] in CEMO-1, MV4-11 and OCI-Ly10 cell lines using GSEA. (**b**) Gene expression of ISGs published by Schoggins JW et al. and Sistigu and colleagues after 48 h of CM-272 treatment in CEMO-1, MV4-11 and OCI-Ly10 cell lines. (**c**) qRT-PCR validation of ISGs in CEMO-1 and MV4-11 cell lines after 48 h of CM-272 treatment. (**d**) qChIP-PCR analysis of ISGs in CEMO-1 and MV4-11 treated for 48 h with 250 and 270 nM of CM-272, respectively. (**e**) Calreticulin exposure determined by FACS analysis of CEMO-1, MV4-11 and OCI-Ly10 cells after 48 h of treatment with GI$_{25}$ and GI$_{50}$ of CM-272. (**f**) HMBG1 secretion determined by ELISA analysis of supernatants of CEMO-1, MV4-11 and OCI-Ly10 cells after 48 h of treatment with GI$_{25}$ and GI$_{50}$ of CM-272. Error bars indicate s.d. from three replicates.

dose-dependent efficacy of CM-272 and that a dose of 2.5 mg kg$^{-1}$ of CM-272 is adequate to demonstrate the positive anti-tumour efficacy. In a second xenogeneic model, $10 \times 10^{6}$ of AML-derived MV4-11 cells were injected i.v. in Rag2$^{-/-}$γc$^{-/-}$ mice, and 14 days latter animals were treated with 2.5 mg kg$^{-1}$ of CM-272 for 28 days. As in ALL cells, CM-272 therapy prolonged OS in mice (median OS for treated versus untreated mice, $78 \pm 12$ days versus $57 \pm 0.9$ days; $P = 0.0005$) (Fig. 4b). We obtained similar results in a second *in vivo* replicate with MV4-11 cells (Supplementary Fig. 11b), without any sign of toxicity (Supplementary Fig. 10h). Finally, $2.5 \times 10^{6}$ cells from the OCI-Ly10 activated B-cell DLBCL cell line were similarly i.v. injected into Rag2$^{-/-}$γc$^{-/-}$ mice. Treatment with CM-272 at the same dose during 8 weeks also prolonged OS of treated mice in comparison to control animals (median OS; $59 \pm 8$ days versus $49 \pm 6$ days; $P = 0.010$) (Fig. 4c). We obtained similar results in a second *in vivo* replicate with OCI-Ly10 cells (Supplementary Fig. 11c), without any sign of toxicity (Supplementary Fig. 10i). Although the effect on lymphoma cells was statistically significant, the effect was less robust than in the case of AML and ALL cells. These results show that CM-272 exerts a potent anti-tumour activity *in vivo* against different types of haematological malignancies by inhibiting the methyltransferase activity of both G9a/GLP and DNMTs. In addition to the information described above, minimal promiscuity versus other SAM-dependent epigenetic enzymes (Supplementary Tables 4a and 4b), further off-target selectivity profiling against other drug targets in cancer (a panel of 97 kinases, Supplementary Tables 12–14) confirmed G9a (and GLP) and DNMTs as primary main targets for CM-272.

## Discussion

The development of epigenetic therapies is increasingly recognized as a highly attractive field for different tumours. Besides already approved HDAC inhibitors and hypomethylating agents, histone lysine methylation represents an emerging therapeutic target in cancer[5]. However, posttranscriptional gene regulation frequently involves several levels of control, suggesting that the combination of drugs modulating different epigenetic mechanisms may represent a more efficient therapeutic approach with respect to one-drug treatment[34]. In this study, we demonstrate as a proof of principle that a single compound simultaneously targeting DNMTs and G9a/GLP methyltransferase activity is significantly active *in vitro* and *in vivo* against different haematological tumours with poor prognosis. The association between G9a with DNMTs[11,35] in the regulation of tumour suppressor genes[36] supports the rationale for targeting both epigenetic mechanisms for cancer treatment. Unlike other recently described KMT inhibitors, our compounds

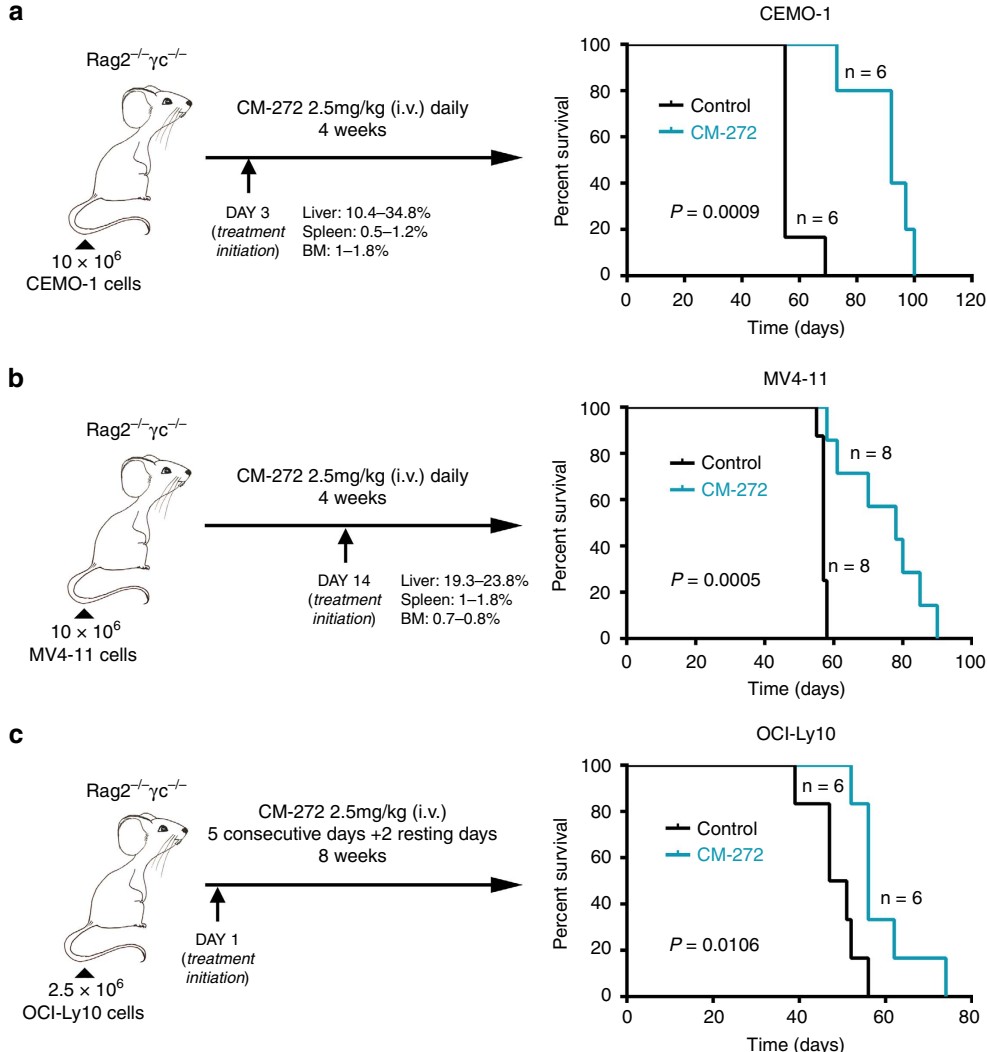

**Figure 4 | CM-272 shows anti-leukaemic effects *in vivo*.** Schematic diagram of *in vivo* CM-272 treatment procedure and Kaplan–Meier survival curves for evaluating the survival time of mice engrafted with ALL-derived CEMO-1 cells (**a**), AML-derived MV4-11 (**b**) and DLBCL-derived OCI-Ly10 (**c**) treated with CM-272. Infiltration levels in liver, spleen and bone marrow are indicated in the figure. Control: saline solution (diluent of CM-272). *P* values assessed by log-rank.

do not compete for SAM[37] but instead simultaneously inhibit binding of G9a (also its closely related protein GLP) and DNMTs to their substrates reversibly, which may explain their selectivity, lack of off-target effects and therapeutic potential in different tumour types.

The discovery of biomarkers that allow monitoring of therapeutic efficacy is an important aim in the development of new drugs. Although a correlation between baseline levels of G9a or H3K9me2 and tumour responses was not observed, our *in vitro* and *in vivo* studies showed that anti-tumour efficacy due to pharmacological inhibition of methyltransferase activity of G9a and DNMTs was associated with global reductions in H3K9me2 and 5mC levels. Accordingly, sequential measurement of H3K9me2 and 5mC levels during CM-272 treatment may potentially be used as biomarkers to assess its therapeutic efficacy.

The results of the transcriptomic analysis performed after treatment with CM-272 consistently suggest a tumour interferon type 1 response with expression of ISGs and induction of immunogenic cell death (ICD) in AML, ALL and DLBCL cells, pointing to a common mechanism of anti-tumour effect. Despite the fact that ICD had not been described as a mechanism of

action for epigenetic drugs, these results might have been at least partially predicted, because recent studies have shown that expression of ISGs is epigenetically regulated by H3K9me2 (ref. 30), supporting the role of G9a inhibition in the activation of type I interferon responses and ICD. Thus, we speculate that the use of immunocompromised mice unable to develop anti-tumour immune responses may have underestimated the efficacy of CM-272 against tumour cells, prompting the evaluation of immune-competent models to explore the full potential therapeutic effects of our compounds[33]. On the basis of recent studies demonstrating that type I interferon responses contribute to the efficacy of chemotherapeutic agents[32], the use of CM-272 in combination with such drugs and/or with immune modulators such as checkpoint inhibitors might also represent an attractive therapeutic strategy.

In summary, CM-272 is a potent novel first-in-class dual reversible inhibitor of G9a (GLP) and DNMTs that prolongs survival in *in vivo* models of haematological malignancies by at least in part inducing immunogenic cell death. These compounds represent a novel approach for targeting cancer safely and efficiently, paving the way for treating a broad series of human tumours with poor prognosis.

## Methods

**G9a and DNMT1 docking.** CM-272 was docked into the co-crystal structures of the G9a-UNC0638-SAH complexes (Protein Data Bank, PDB, entry 3RJW) and DNMT1 (PDB entry 4DA4) by using Gold software.

**Compound synthesis of CM-272 and CM-579 and CM-1021.** A solution of 2-methoxy-5-nitro-phenol in THF was reacted with 3-pyrrolidin-1-ylpropan-1-ol in the presence of PPh3 and DEAD to yield 1-[3-(2-methoxy-5-nitro-phenoxy)propyl]pyrrolidine (compound **1**), which was later hydrogenated to yield 4-methoxy-3-(3-pyrrolidin-1-ylpropoxy)aniline (compound **2**). Then, 2,4-dichloro-6-methoxy-7-(3-pyrrolidin-1-ylpropoxy) quinoline (compound **3**) was obtained by reacting compound **2** with malonic acid in POCl$_3$ to obtain the quinoline core. Suzuki coupling of compound **3** with 4,4,5,5-tetramethyl-2-(5-methyl-2-furyl)-1,3,2-dioxaborolane afforded derivatization at the 2-position to yield 4-chloro-6-methoxy-2-(5-methyl-2-furyl)-7-(3-pyrrolidin-1-ylpropoxy)quinoline (compound **4**). Finally, compound **4** was cross-coupled with the corresponding amine (1-methylpiperidin-4-amine for CM-272 and 1-(methylpiperidin-4-yl)methanamine for CM-579) under Pd$_2$(dba)$_3$ and BINAP catalysis to yield target compounds CM-272 (6-methoxy-2-(5-methyl-2-furyl)-N-(1-methyl-4-piperidyl)-7-(3-pyrrolidin-1-ylpropoxy)quinolin-4-amine) and CM-579 (6-methoxy-2-(5-methyl-2-furyl)-N-[(1-methyl-4-piperidyl)methyl]-7-(3-pyrrolidin-1-ylpropoxy)quinolin-4-amine). CM-272 m.p. 180–181 °C. $^1$H NMR (MeOD, 400 MHz): δ 7.81 (s, 1H), 7.58 (d, $J = 2.8$ Hz 1H), 7.50 (s, 1H), 7.09 (s, 1H), 6.44 (s, 1H), 4.35 (m, 4H), 4.04 (s, 3H), 3.83 (m, 2H), 3.72 (m, 2H), 3.51 (m, 2H), 3.27 (m, 1H), 3.17 (m, 2H), 2.95 (m, 3H), 2.51 (s, 3H), 2.40 (m, 4H), 2.21 (m, 6H). $^{13}$C NMR (DMSO-$d_6$, 100 MHz): δ 157, 153, 152.9, 148.7, 144, 139.6, 134.6, 116.5, 110.3, 110, 102.7, 101.2, 92.1, 66.1, 56.6, 53.2 (2C), 52.5 (2C), 51.4, 47.7, 42.7, 28.5 (2C), 25, 22.6 (2C), 13.6. HRMS $m/z$: $[(M + H)]^+$ calcd. for C$_{28}$H$_{38}$N$_4$O$_3$: 479.3022; found, 479.3060. CM-579 m.p. 34–35 °C. $^1$H NMR (MeOD, 400 MHz): δ 7.73 (s, 1H), 7.55 (d, $J = 2.4$ Hz, 1H), 7.48 (s, 1H), 7 (s, 1H), 6.42 (d, $J = 2.4$ Hz, 1H), 4.35 (m, 2H), 4.03 (s, 3H), 3.82 (m, 2H), 3.62-3.56 (m, 4H), 3.51-3.47 (m, 2H), 3.17 (m, 2H), 3.03 (m, 2H), 2.86 (m, 3H), 2.50 (s, 3H), 2.39-2.36 (m, 2H), 2.21–2.06 (m, 7H), 1.65 (m, 2H). $^{13}$C NMR (DMSO-$d_6$, 100 MHz): δ 156.8, 153.9, 153, 148.7, 143.9, 139.4, 134.4, 116.3, 110.2, 109.9, 102.4, 101.2, 91.9, 66.1, 56.3, 53.2 (2C), 53 (2C), 51.3, 47.1, 42.6, 32.1, 26.9 (2C), 25, 22.6 (2C), 13.6. HRMS $m/z$: $[(M + H)]^+$ calcd. for C$_{29}$H$_{40}$N$_4$O$_3$, 493.3179; found, 493.3178. Reaction of compound **4** with 4,4,5,5-tetramethyl-2-(4,4,5,5-tetramethyl-1,3,2-dioxaborolan-2-yl)-1,3,2-dioxaborolane and posterior hydrolysis afforded 6-methoxy-2-(5-methyl-2-furyl)-7-(3-pyrrolidin-1-ylpropoxy)quinolin-4-ol (compound **6**), which was transformed into CM-1021 (6-methoxy-2-(5-methyl-2-furyl)-4-[(1-methyl-4-piperidyl)oxy]-7-(3-pyrrolidin-1-ylpropoxy)quinoline) by nucleophilic substitution with *tert*-butyl 4-methylsulfonyloxypiperidine-1-carboxylate and posterior methylation. CM-1021 Yellow oil. $^1$H NMR (MeOD, 400 MHz): δ 7.78 (s, 1H), 7.69–7.57 (m, 3H), 6.51 (d, $J = 2.8$ Hz, 1H), 4.41–4.38 (m, 2H), 4.11–4.05 (m, 3H), 3.87–3.73 (m, 3H), 3.52–3.46 (m, 5H), 3.18–3.13 (m, 2H), 2.61–2.57 (m, 3H), 2.56 (s, 3H), 2.42–2.39 (m, 4H), 2.27–2.18 (m, 3H), 2.10–2.07 (m, 2H). $^{13}$C NMR (DMSO-$d_6$, 100 MHz): δ 153.6, 149.6, 120.9, 118, 115, 114.4, 114.3, 112.1, 110.3, 101.3, 100.7, 96.9, 96.8, 66.3, 56.2, 56, 53.3 (2C), 51.4 (2C), 48.8, 42.5, 27.8, 26.4, 25, 22.6 (2C), 13.7. HRMS $m/z$: $[(M + H)]^+$ calcd. for C$_{28}$H$_{37}$N$_3$O$_4$, 480.2862; found, 480.2853. Further details on compound characterization, reagents and conditions can be found in the Supplementary Information.

**G9a and DNMT1 enzyme activity assays.** G9a and DNMT1 activities were measured using a time-resolved fluorescence energy transfer (TR-FRET). For G9a, TR-FRET is observed when biotinylated histone monomethyl-H3K9 peptide is incubated with cryptate-labelled anti-dimethyl-histone H3K9 antibody (CisBio Cat # 61KB2KAE) and streptavidin XL665 (CisBio Cat # 610SAXLA) after enzymatic reaction of G9a. For DNMT1, TR-FRET is observed when antibody specific to S-adenosylhomocysteine labelled with Lumi4-Tb (donor) is incubated with d2-labelled S-adenosylhomocysteine (acceptor), using the EPIgeneous methyltransferase assay (CisBio Cat # 62SAHPEB). Details are provided in Supplementary Information.

The radioligand binding assay against G9a, DNMT1 and GLP was performed by Reaction Biology Corporation (http://www.reactionbiology.com).

**Epigenetics selectivity panel.** Selectivity of CM-272 and CM-579 against 37 epigenetic enzyme targets including Bromodomain-containing enzymes (ATAD2A, ATAD2B, BAZ2B, BRD1, BRD2(BD1 + BD2), BRD4(BD1 + BD2), BRDT(BD1), CREBBP, TRIM24, TAF1), Histone methyltransferases (EZH1, EZH2, GLP, PRMT1, PRMT3, PRMT4, PRMT5, PRMT6, PRMT8, SETD2, SET7/9, SUV39H1, SUV39H2 and MLL-WARD), DNA methyltransferases (DNMT3A and DNMT3B) and histone demethylase (JMJD2A, JMJD2B, JMJD2C, JMJD2D, JMJD2E, JMJD3, JMJD1A, LSD1, Jarid1A, Jarid1B and Jarid1C) was performed by BPS Bioscience (http://www.bpsbioscience.com/index.ph).

**HDAC1 and HDAC6 enzyme activity assays.** HDAC1 and HDAC6 enzyme activities were measured with a specific fluorescence-labelled substrate (BPS Biosciences, Cat # 50037), containing an acetylated lysine side chain, after its deacetylation by HDACs. Details are provided in Supplementary Information.

**ADME profiling.** The following ADME studies: CYP inhibition on five human cytochrome P450s (1A2, 2C9, 2C19, 2D6 and 3A4 at 10 μM) in human liver microsomes, plasma protein binding, kinetic solubility, Pampa permeability and human and mouse liver microsomal stability were performed by Wuxi (http://www.wuxi.com/). Details are provided in Supplementary Information.

**hERG blockade assay.** The effect of the compound on hERG potassium channels was determined using the PredictorTM hERG fluorescence polarization commercial assay kit. Details are provided in Supplementary Information.

**Cytotoxicity in THLE-2 cells and PBMCs.** Cell viability was determined by measuring the concentration of cellular adenosine triphosphate (ATP) using the ATP1Step Kit as described by the manufacturer (Perkin-Elmer). PBMCs were isolated following the regular density gradient centrifugation procedure with Ficoll. Details are provided in Supplementary Information.

**PK study of CM-272 and CM-579 in plasma samples.** CM-272 and CM-579 were measured in plasma samples using a Xevo-TQ MS triple quadrupole mass spectrometer with an electrospray ionization (ESI) source and an Acquity UPLC (Waters, Manchester, UK). CM-272 and CM-579 solutions were prepared by dissolving the solid in saline. A drug dosage of 1 mg kg$^{-1}$ or 2.5 mg kg$^{-1}$ (CM-272) or 1 mg kg$^{-1}$ (CM-579) was administered as a single intravenous injection. Blood was collected at predetermined times over 24 h post injection (0.25, 2, 4, 6, 8 and 24 h for CM-272) and (0.25, 1, 2, 4 and 8 h for CM-579). Chromatographic separation was performed by gradient elution at 0.6 ml min$^{-1}$ using an Acquity UPLC BEH C18 column (50 × 2.1 mm, 1.7 μm particle size; Waters). The PK parameters were obtained by fitting the blood concentration-time data to a non-compartmental model with the WinNonlin software (Pharsight, Mountain View, CA, USA). Details are provided in Supplementary Information.

**Kinase selectivity profiling.** The selectivity profiling of CM-272 against a selected panel of 97 kinases distributed through the kinome (out of which 90 are non-mutant kinases) was performed at DiscoverRx (http://www.discoverx.com/home) using the KINOMEscan screening platform at a test concentration of 10 μM.

**Direct binding analysis.** MicroScale Thermophoresis (MST) was performed to quantify biomolecular interactions between CM-579 and DNMT1 (full length). The MST analysis was performed using the Monolith NT.115 instrument (NanoTemper Technologies).

**Cell proliferation and apoptosis and cell cycle assays.** Cell proliferation was analysed using the CellTiter 96 Aqueous One Solution Cell Proliferation Assay following the manufacturer's instructions (Promega, Madison, WI, USA). The GI$_{50}$ values of the different compounds were determined using non-linear regression plots with the GraphPad Prism v5 software. Cell cycle assay is detailed in Supplementary Information. Apoptosis was analysed by flow cytometry using the FITC Annexin V Apoptosis Detection Kit I (BD Pharmingen) following the manufacturer's instructions. Details are provided in Supplementary Information.

**Combination assay.** For the calculation of combination index (CI) values, OCI-AML-2 growth inhibition was determined at multiple concentrations of G9a inhibitor (A-366) (12.5, 25, 50 and 100 μM) in combination with varied concentrations of decitabine (12.5, 25, 50 and 100 μM) as described in Supplementary Information. The resulting data were analysed according to the method described by Chou (Calcusyn software, Biosoft). CI was used to determine whether the effect of drug combinations were synergistic, additive or antagonistic. Synergy, additivity and antagonism were defined by a CI less than one, one and greater than one, respectively.

**Western blot.** Histone extraction was performed and analysed as described in Supplementary Information. Images have been cropped for presentation. Full size images are presented in Supplementary Figs 13 and 14.

**Dot blot.** Genomic DNA was extracted using a DNA kit (Nucleo Spin Tissue, Cat # 74095250, Macherey–Nagel) following the manufacturer's instructions. Overall, 500 ng of genomic DNA was loaded onto a nitrocellulose membrane (Amersham Hybond_N +, RPN203B, GE Healthcare) using the Bio-Dot microfiltration apparatus (Cat # 170-6545, BioRad) following the manufacturer's instructions. Dot blot of 5-methylcytosine was performed as described in Supplementary Information.

**Bioinformatics analyses.** RNA-Seq and expression arrays methods are detailed in Supplementary Information. These data can be downloaded from Gene Expression Omnibus public functional genomics data repository under the accession number of super serie GSE78932.

**qRT-PCR.** qRT-PCR was performed in a 7300 Real-Time PCR System (Applied Biosystems), using 20 ng of cDNA in 2 µl, 1 µl of each primer at the concentration specified in Supplementary Information, 6 µl of SYBR Green PCR Master Mix 2X (Cat # 4334973, Applied Biosystems) in 12 µl reaction volume. The following programme conditions were applied for qRT-PCR running: 50 °C for 2 min, 95 °C for 60 s following by 45 cycles at 95 °C for 15 s and 60 °C for 60 s; melting programme, one cycle at 95 °C for 15 s, 40 °C for 60 s and 95 °C for 15 s. The relative expression of each gene was quantified by the log $2^{(-\Delta\Delta Ct)}$ method using the gene *GUS* as an endogenous control.

**Quantitative-chromatin immunoprecipitation.** Quantitative-chromatin immunoprecipitation (Q-ChIP) was performed in a 7300 Real-Time PCR System (Applied Biosystems) as described in Supplementary Information. The percentage of H3K9me2 of each gene was quantified calculating [($2^{(-\Delta\Delta Ct)}$ CM-272 sample/ $2^{(-\Delta\Delta Ct)}$ control) × 100].

**DNA methylation analysis.** DNA methylation status of *CDKN2A* and *POU4F2* promoters was analysed by pyrosequencing techniques. Genomic DNA extraction and bisulfite modification were performed as described in Supplementary Information. The pyrosequencing reactions were performed using the PyromarkTM ID (Biotage) as described in Supplementary Information and sequence analysis was performed using the PyroQ-CpG analysis software (Biotage).

**Determination of cell surface-exposed calreticulin.** Cells were incubated with a CRT-specific antibody (Abcam) and stained TO-PRO-3 (Life Technologies) and analysed by means of a FACSCalibur cytofluorometer (BD Biosciences) as described in Supplementary Information. First line statistical analyses were performed by using the CellQuest software (BD Biosciences), upon gating on TO-PRO-3 negative events characterized by normal forward and side scatter (living cells).

**Determination of extracellular HMGB1 concentrations.** Extracellular HMGB1 from 48 h cell culture supernatants was quantified by means of the HMGB1 ELISA Kit II (Shino Test), following the manufacturer's instructions.

***In vivo* experiments.** All animal studies had previous approval from the Animal Care and Ethics Committee of the University of Navarra, whereas experiments that used patient samples were approved by the Human Research Ethics Committees of University of Navarra.

The human ALL CEMO-1 (control group with saline solution $n = 6$; treated group with CM-272, $n = 6$), AML MV4-11 (control group with saline solution $n = 8$; treated group with CM-272 $n = 8$) and DLBCL OCI-Ly10 (control group with saline solution $n = 6$; treated group with CM-272 $n = 6$) xenograft mice models were generated by i.v. injection of cells diluted in 100 µl of saline solution in the tail vein of a 6–8-week-old female BALB/cA − Rag2$^{-/-}$γc$^{-/-}$ mice as described in Supplementary Information. CM-272 administration was detailed in Supplementary Information. Statistical results were calculated using the statistical software medcalc.

**CM-272 toxicity assay: haematological and liver parameters.** After treating Rag2$^{-/-}$γc$^{-/-}$ mice with daily i.v. 2.5 mg kg$^{-1}$ of CM-272 during 4 weeks, followed by a 7 days washout period, haematological and liver parameters were measured as described in Supplementary Information.

**Statistical analysis.** Data are expressed as mean ± s.d. To compute *P* values for bar graphs, a two-tailed Mann–Whitney *U* or Student's *t* test were used. In survival curves, significance was calculated using log-rank analysis. GraphPad Prism 4.0 was used to carry out all statistical analyses.

**Data availability.** The authors declare that all data supporting the findings of this study are available within the article and its Supplementary Materials, or from the corresponding authors on request. The RNA-Seq and expression arrays methods data can be downloaded from Gene Expression Omnibus public functional genomics data repository under the accession number of super serie GSE78932.

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

## Acknowledgements

We particularly acknowledge the Biobank of the University of Navarra for its collaboration. We thank Dr Edorta Martínez de Marigorta and Dr Francisco Palacios from Departamento de Química Orgánica I, Facultad de Farmacia, Universidad del Pais Vasco for 13C NMR determination and Angel Irigoyen Barrio and Dr Ana Romo Hualde, from University of Navarra, for HRMS determination. Dr. Irene de Miguel Turrullols from Small Molecule Discovery Platform, CIMA, University of Navarra is acknowledged for NMR data interpretation. This work was funded by grants from Instituto de Salud Carlos III (ISCIII) PI10/01691, PI13/01469, PI14/01867, PI10/2983, TRASCAN (EPICA), CIBERONC, cofinanciacion FEDER, RTICC RD12/0036/0068, Fundació La Marató de TV3 (20132130-31-32) and 'Fundación Fuentes Dutor'. B.P. is supported by a Sara Borrell fellowship CD13/00340 and X.A. is a Marie Curie researcher under contract 'LincMHeM-330598'.

## Author contributions

Conception and design: E.S.J.-E., X.A., O.R., J.O. and F.P. Development of methodology: E.S.J.-E., X.A., O.R., A.V.-Z., J.A.S.-A., A.U., B.P., N.C., M.G.F.-B. and J.O. Acquisition of data: E.S.J.-E., X.A., O.R., A.V.-Z., J.A.S.-A., E.M., A.U., S.R., B.P., R.M.A., F.-X.O., P.S., C.M.S., R.C.-O., N.C., M.G.F.- B., J.R.R.-M., M.J.G.-B. Analysis and interpretation of data: E.S.J.-E., X.A., O.R., A.V.-Z, S.R., B.P., V.S., A.E.-H.de M., F.-X.O., C.M.S., R.C.-O., J.I.M.-S., G.C., J.J.-L., M.A.A., J.A.M.-C., J.O. and F.P. Writing and/or revision of the manuscript: E.S.J.-E., X.A., O.R., J.I.M.-S., R.C.-O., J.A.M.-C., J.O. and F.P. Study supervision: X.A., J.O. and F.P.

## Additional information

**Competing interests:** The authors declare no competing financial interests.

