## [Peer Review File · Nature Communications]

Reviewers' comments:

Reviewer #1 (Remarks to the Author):

Thanks for the opportunity to review this interesting manuscript. The authors generate a novel G9a and Dnmt inhibitor and validate this in vitro and in vivo. The chemical biology approach to generate the inhibitor and screen with TR-FRET is noted, 2 tool compounds are selected for future work. This is examined in cell lines (cancer vs. normal tissues) and shows activity in nanomolar range associated with reductions in H3K9me2 and 5mc. A modest effect on apoptosis and cell cycle is noted. Mechanistically, there is association with t1 interferon expressed genes, these are validated with QPCR and reduced H3K9me2 at the promoters. The hypothesis is that cell-autonomous immunological death is induced. PK studies are performed in normal mice and finally, cell line xenograft experiments show prolongation of survival in the same cell lines as used throughout (CEMO, MV4;11 and OCI-Ly10). The DLBCL line has the least sensitivity throughout.

The work is original and of significant to many researchers in cancer biology. Some major points would need to be addressed before publication.

Major points:

1. Some clarifications are required regarding quality and reproducibility of data, in particular Fig 4. Specifically, please specify the disease burden for each group separately and show graphically (not pooled infiltration levels); how many replicate experiments were performed (it appears to be a single in vivo experiment per cell line currently, multiple replicates are required for such a figure); please list the control (Saline based on supp data) and diluent, please provide additional toxicity rather than just body weight (blood counts would be useful).

Global H3K9me2 is shown to be lower in the treated mice livers - is this leukaemia cells purified, or total liver extract?

The effect of the drug in vivo is variable, and at times, quite modest (e.g. in the DLBCL experiments).

It's not clear why different treatment schedules were used in these experiments for different cell lines.

2. Mechanism: IFN gene regulation is shown and is associated with reduced H3K9me2. Have the authors looked at other genes that do not have differential expression? What is the status of H3K9me2 at these gene promoters? i.e. is the H3K9me2 effect global vs. specific for differentially regulated genes.

Generally speaking - gene regulation is shown, but this is not mechanistically linked to cell death in vitro or in vivo. Currently, it remains an association.

3. What does cell autonomous immunological death mean?

4. If the mechanism of death is immunological, please explain why prolonged survival of mice is seen in Rag/gc^{-/-} mice? Have they examined survival in an immune-competent tumour model?

I have no concerns regarding referencing, quality of work otherwise. Multiple in vivo replicates from each tumour model are absolutely required for robust data and reproducibility.

Minor point - table S2 has an error in the uM text.

Reviewer #2 (Remarks to the Author):

The study investigates the dual inhibition of the methyltransferases G9a and DNMT1, primarily using novel bispecific inhibitors. The authors show a wealth of in vitro and cell-based data, providing evidence for the on-target effects on these inhibitors on AML cell viability.

Although inhibitors of the individual enzymes have been described, this is the first report of a bi-specific inhibitor, which has enabled an in-depth analysis of the combined effects of inhibiting both enzymes.

Overall the paper is very clearly written and the data is presented well.

I applaud the level of detail given on the methods used in the study - this made it very easy to check that the methods used were appropriate and will help other groups who may wish to reproduce some of the experiments.

There is scope for some improvement:

1) The methyltransferase inhibitors identified in the study are clearly related to known inhibitors of G9a such as UNC0638. However, in contrast to the previous compounds, CM-272 also inhibits DNMT1. Based on the figures presented in the manuscript, it is difficult to see whether the binding sites for this compound on G9a/DNMT1 are related, and so the basis for dual specificity is unclear. This is because the views shown in Figure 1C and D are very different (as indicated by the different orientations of SAH). The authors should clarify whether the binding sites are structurally related, and include a superposition of the models. I also recommend that the authors include a description of the structural features that underpin the selectivity of the compound compared to related enzymes.

2) The study is based on the hypothesis that dual inhibition of G9a and DNMT1 is a potent combination in killing cancer cells. Early in the text, a combination study is reported using inhibitors of DNMT1 (Decitabine) and G9a (A-366) on OCI-AML-2 cell line. This shows synergy, but it is difficult to compare these results with later experiments (using the bispecific inhibitors) because the data that underpins the CI calculations is not shown. I request that this data is included in a revised manuscript.

3) There remains a lingering doubt in my mind whether the transcriptional changes and cell death induced by CM272 is due to the bi-specific inhibition of G9a/DNMT1 or whether the inhibition of other enzymes (such as PRMT1,6 and/or GLP) might also contribute. This possibility is dismissed too easily by the authors (final sentence of the Results). To address this, I suggest the authors carry out studies using genetic knockdown (e.g. RNAi) in combination with chemical inhibitors in a selected set of cell lines from their panel to determine whether other epigenetic targets might also contribute to immunogenic cell death.

Reviewer #3 (Remarks to the Author):

The manuscript by Prosper, et al describes the discovery of reversible dual inhibitors of G9a and DNMTs and characterization of these inhibitors in cellular and in vivo models of hematological malignancies. There are a number of major issues noted below. In particular, the lead compounds CM-272 and CM-579 have not been convincingly characterized as true inhibitors of DNMTs. Therefore, it is questionable that the observed effects in cell-based and in vivo studies are truly due to pharmacological inhibition of G9a and DNMTs. In light of these major concerns, this manuscript is not suitable for publication in a high impact journal such as Nature Communications.

Major issues:

1. Characterization of CM-272 and CM-579 as inhibitors of DNMTs (DNMT1, DNMT3A and DNMT3B) is rather weak, far from convincing. These compounds showed inhibitory activities in only two biochemical assays (TR-FRET and tritiated SAM). Direct binding of these inhibitors to DNMT1,

DNMT3A and DNMT3B should be established by biophysical methods such as ITC and SPR, but is missing. The claim that CM-272 and CM-579 bind to the DNA binding site of DNMT1 is questionable. The MOA studies showed that the compounds are competitive with a DNA substrate, but this result does not tell where the inhibitors bind. Docking studies are not sufficient to establish the binding mode. Either an X-ray cocrystal structure or an NMR structure is needed to convincingly demonstrate that these inhibitors bind to the DNA binding site of DNMTs. A crystal or NMR structure of DNMT1 or DNMT3A in complex with one of the lead compounds would also convincingly demonstrate that the compounds are true inhibitors of DNMTs.

2. It is surprising that authors have not conducted cellular studies by knocking down both G9a and DNMTs via shRNA or siRNA, or introducing catalytically inactive G9a and DNMTs via CRISPR/Cas9. This genetic approach, which is complementary to the pharmacological approach, should be performed to support the conclusion of the manuscript.

3. The toxicity of the lead compounds is also a major concern. CM-272 has an LC50 of 1.8 μ M in THLE-2 (a non-tumoral hepatic cell line) and LC50 of 1.9 μ M in PBMCs. CM-579 displayed similar LC50s. It appears that these inhibitors are generally toxic (Table S9). At best, they have a very small therapeutic window (\sim 1 log unit), which is insufficient. The maximum tolerated dose for CM-272 and CM-579 in mice is 2.5 mg/kg and $<$ 1 mg/kg, again suggesting that these inhibitors are toxic. CM-272 also inhibited multiple kinases (Table S16).

4. PK/PD relationship needs to be established for the in vivo efficacy studies. Although H3K9me2 and 5mC levels were assessed at one time point (Fig. S4F), drug levels should be measured for the treated animals to correlate with the reduction of H3K9me2 and 5mC and with phenotypic effects.

5. Dose-dependent in vivo efficacy should be established, but is missing. The overall survival benefit is modest in all 3 models.

Minor concerns:

1. CM-272 displayed high potency for EHMT1 (GLP) (IC50 = 2 nM). Most of cellular results in the manuscript were generated using CM-272 and all in vivo results were generated using CM-272. It is not clear why authors ignored CM-272's EHMT1 inhibitory activity. Potential contributions from inhibition of EHMT1 cannot be ruled out in most of the cellular and in vivo studies they performed.

2. CM-272 and CM-579 are structurally similar to previously reported inhibitors of G9a and EHMT1. They are likely true inhibitors of G9a and EHMT1. Nevertheless, confirmation using a biophysical method such as ITC or SPR should be performed. A crystal or NMR structure of G9a or EHMT1 in complex with CM-272 would convincingly demonstrate that the inhibitor occupies the substrate-binding site.

3. The effect of CM-272 on other common histone marks (in addition to H3K9me2, H3K9me3 and H3K27me3) should be evaluated.

Response to the Reviewers' comments (manuscript NCOMMS-16-14716 by E. San José-Enériz *et al.*)

We would like to thank the reviewers for their thorough revision, constructive criticisms and suggestions on our manuscript. We believe that this revised version of our study has clearly improved following the reviewers' advice.

Below you can find the detailed answers to all the issues raised by the distinguished reviewers.

Reviewer #1 (Remarks to the Author):

Thanks for the opportunity to review this interesting manuscript. The authors generate a novel G9a and Dnmt inhibitor and validate this *in vitro* and *in vivo*. The chemical biology approach to generate the inhibitor and screen with TR-FRET is noted, 2 tool compounds are selected for future work. This is examined in cell lines (cancer vs. normal tissues) and shows activity in nanomolar range associated with reductions in H3K9me2 and 5mc. A modest effect on apoptosis and cell cycle is noted. Mechanistically, there is association with t1 interferon expressed genes, these are validated with QPCR and reduced H3K9me2 at the promoters. The hypothesis is that cell-autonomous immunological death is induced. PK studies are performed in normal mice and finally, cell line xenograft experiments show prolongation of survival in the same cell lines as used throughout (CEMO, MV4;11 and OCI-Ly10). The DLBCL line has the least sensitivity throughout.

The work is original and of significant to many researchers in cancer biology. Some major points would need to be addressed before publication.

Major points:

1. Some clarifications are required regarding quality and reproducibility of data, in particular Fig 4. Specifically, please specify the disease burden for each group separately and show graphically (not pooled infiltration levels); how many replicate experiments were performed (it appears to be a single *in vivo* experiment per cell line currently, multiple replicates are required for such a figure); please list the control (Saline based on supp data) and diluent, please provide additional toxicity rather than just body weight (blood counts would be useful).

Global H3K9me2 is shown to be lower in the treated mice livers - is this leukaemia cells purified, or total liver extract? The effect of the drug *in vivo* is variable, and at times, quite modest (e.g. in the DLBCL experiments). It's not clear why different treatment schedules were used in these experiments for different cell lines.

We thank the reviewer for his/her comments. Indeed the experiments shown in the original version of the manuscript represent one experiment with at least 6 animals per group. However, two complete *in vivo* replicates have been performed per cell line with very similar results in each one of the three models. These results have been included in the revised version of the manuscript (Figure S10, shown below).

Regarding disease burden in each model, the infiltration in the liver, spleen and bone marrow was calculated in a group of animals in which CEMO-1 and MV4-11 cells were transplanted to establish the animal models but did not receive any therapy. The numbers represent the mean infiltration at day 3 and 14, when treatment was initiated. Assessment of liver, spleen and marrow in each animal during treatment is not feasible without sacrificing the mice.

Figure S10. CM-272 shows anti-leukemic effects in vivo. Kaplan-Meier survival curves for evaluating the survival time of mice engrafted with ALL derived CEMO-1 cells (A), AML derived MV4-11 (B) and DLBCL derived OCI-Ly10 (C) treated with CM-272. Control: Saline Solution (diluent of CM-272).

As suggested by the reviewer, we have included in the figure legends (Figure 4 and S10) and in the revised version of the manuscript that saline solution was used as control and as diluent of CM-272. Changes are included in the results section (page 11), and in supplementary materials (page 18).

Hematological and liver toxicity of CM-272 was examined in healthy Rag2^{-/-}γc^{-/-} mice and the results were included in the original version of the manuscript and also as supplementary information (Results section page 11 and page 12; Supplementary Materials Page 18, Figure S9). In healthy animals, no evidence of weight loss (Figure S9B), hematological toxicity (Figure S9C) or liver toxicity were observed (Figures S9D and S9E) in comparison with control treated animals. Nevertheless, following the recommendation from Reviewer 1, we have performed new experiments to assess haematological toxicity in Rag2^{-/-}γc^{-/-} mice transplanted with CEMO-1 ALL cells. BALB/cA- Rag2^{-/-}γc^{-/-} mice between 6 and 8 weeks of age were injected through the tail vein with 10x10⁶ CEMO-1 cells diluted in 100 μL of saline solution. Treatment with 2.5 mg/kg of CM-272 or saline was initiated at day 3 for 28 consecutive days followed by a 7 days washout period. As shown in figure below, no significant differences in weight or hematological parameters were observed between CM-272 treated animals and controls. These results have been included in the revised version of the manuscript (Results section, page 11-12 and Figure S11).

Figure S11. Body weight and hematological parameters in *in vivo* ALL model. Mean \pm SEM body weight and hematological parameters in CEMO-1 mice treated with vehicle or 2.5mg/kg of CM-272 daily for 4 weeks followed by a 7 days washout period. WBC: white blood cells; RBC: red blood cells; PLT: platelet count. Black: Control group with saline solution; Blue: CM-272 treatment

Assessment of global H3K9me2 was measured in the extract from total liver. However, at the time of sacrificed we observed huge livers in mice, with the majority of their area constituted by human cells. Livers were homogenized and subjected to flow cytometry showing that 60-80% of these cells were human (hCD45+). These results have included in the revised version of the manuscript (Results section, page 11-12).

The effect of CM-272 is clearly different in ALL, AML and DLBCL (less active in lymphoma), in fact the different sensitivity in each disease is the reason for the different schedules. The schedule was the same for AML and ALL models and the response in both groups was very significant with an increase in survival. Although understanding the differences in sensitivity is certainly interesting, the aim of the current study was to demonstrate that the dual inhibition with a new chemical series of the methyltransferase activity of G9a and DNMTs is an interesting approach for different human tumors. Future studies should focus on understanding mechanisms of resistance in different tumors.

2. Mechanism: IFN gene regulation is shown and is associated with reduced H3K9me2. Have the authors looked at other genes that do not have differential expression? What is the status of H3K9me2 at these gene promoters? i.e. is the H3K9me2 effect global vs. specific for differentially regulated genes. Generally speaking - gene regulation is shown, but this is not mechanistically linked to cell death *in vitro* or *in vivo*. Currently, it remains an association.

In the initial analysis we decided to focus on expression and regulation of genes whose expression changed after treatment with CM-272 as a means to understand the potential mechanisms of action. However, following the reviewer's suggestion, we carefully examined the expression and histone modification (H3K9me2) of genes that did not change after CM-272 treatment in leukemia cell lines. We focus on *PRDM5*, *NDNF* and *Linc582* as its expression did not change after treatment of CEMO-1 cells with CM-272 according to the arrays and RNA-Seq results. New experiments were repeated by treating CEMO-1 cells with 250nM of CM-272 for 48 hours after which Q-RT-PCR and Q-ChIP of H3K9me2. As shown in the figure below the expression of *PRDM5*, *NDNF* and *Linc582* did not change after CM-272 treatment nor the levels of H3K9me2 in the promoter region of these genes. These results indicate that changes in H3K9me2 after CM-272 treatment are not a global effect and that epigenetic modulation after this treatment occurs in specific genomic regions and genes of leukemic cells.

These results have been included in the revised version of the manuscript with a new supplemental figure (Figure S8), in the results section (page 10) and supplementary materials section (page15-17).

Figure S8. mRNA expression and H3K9me2 levels after CM-272 treatment of leukemic cells. (B) Q-RT-PCR validation of *PRDM5*, *NDNF* and *Linc582* in CEMO-1 cell line treated for 48h with 250nM of CM-272. **(C)** Q-ChIP-PCR analysis of *PRDM5*, *NDNF* and *Linc582* in CEMO-1 cell line treated for 48h with 250nM of CM-272.

We also agree with the reviewer that our results do not demonstrate a mechanistic link between gene expression changes and leukemic cell death *in vitro* and *in vivo* but only an interesting association. In fact, while most changes in gene expression were related with Interferon pathway and immunogenic cell death activation, other pathways involved in cell proliferation, cell death, DNA repair and cell metabolism including TP53 and MYC related pathways were deregulated (see figure below). However, we decided to focus on the IFN pathway for 2 main reasons 1) recent studies suggest the importance of IFN pathway in tumor surveillance and the epigenetic regulation of this pathway; 2) additional studies performed with CM-272 in different tumor models such as myeloma, bladder cancer and hepatocellular carcinoma consistently have shown that cell death is associated with upregulation of IFN related genes which are regulated by specific epigenetic modulation through H3K9me2, suggesting a general mechanism of induced tumor cell death.

Ingenuity Pathway Analysis of transcriptome changes after CM-272 treatment in CEMO-1, MV4-11 and OCI-Ly10 cell lines. CM-272 treatment resulted in significantly altered levels of mRNA of genes enriched in disease and function analysis (**A**, **B**) and upstream regulators (**C**). **A**) The Ingenuity Pathway Analysis (IPA) disease/function analysis confirmed that genes altered after CM-272 treatment corresponded to cell cycle, DNA repair and cell death among others. **B**) Heatmap with the IPA results of Disease/function. **C**) Heatmap with the IPA results of cell specific pathways. Potential upstream regulators identified after CM-272 treatment included TP53, MYC and interferon related genes. The calculated z-scores indicates a pathway with genes exhibiting overall increased mRNA level (orange) of decreased mRNA level (blue).

For the sake of space this new figure has not been included in the revised version of the manuscript but we have included these results in the revised version of the manuscript (page 9). Should the reviewer deemed it necessary, we would be happy to include the figure as a new supplementary figure.

3. What does cell autonomous immunological death mean?

In addition to their direct effect on tumor cell growth, anti-cancer therapies could operate by directly modulating immune effector cells rather than improving the immunogenic potential of tumors. We refer to “cell autonomous immunological death” when cancer cell is induced to die in a way able to drive the induction of an anti-tumor immune response. As described by others, the death of cancer cells exposed to therapy can be a relatively “silent” process in terms of immunogenicity. However, some drugs can induce the expression of the so called damage associated molecular patterns (DAMPs) such as Calreticulin and HMGB1 (Krysko O et al. Cell Death and Disease. 2013; 4, e631), which are endogenous molecules that are concealed intracellularly in normal conditions, but are exposed or released upon stress, injury, cell death, thereby becoming able to bind cognate receptors on immune cells (Green DR et al. Nat Rev Immunol. 2009; 9(5):353-63). “Cell autonomous Immunological cell death” includes the release of DAMPs, owing to the stimulation of distinct danger signaling pathways occurring in synchrony with cell death signaling (Garg AD et al. P Int J Dev Biol. 2015; 59(1-3):131-40.). We believe that the increase in Calreticulin and HMGB1 suggest that this mechanism is actually involved in CM-272 induced cell death.

4. If the mechanism of death is immunological, please explain why prolonged survival of mice is seen in Rag/gc-/- mice? Have they examined survival in an immune-competent tumour model?

This is a very important issue rightfully raised by the reviewer and that we partially discussed in the original version of the manuscript. As mentioned above, CM-272 can exert its anti-tumoral effect by two ways, (i) by inducing the direct death of the tumor cells and (ii) as an added value, by the expression of DAMPs able to drive the induction of an anti-tumor immune response. The direct anti-tumor effect of CM-272 is clearly demonstrated by the *in vitro* experiments where the immune mediated effect plays no role, so based on these results we expected a direct effect in the *in vivo* model despite the lack of immune system.

In accordance with the reviewer’s suggestion, we have performed a number of experiments aimed to identify an immunocompetent mouse model in which we could test the second immune mediated mechanism of CM-272. We analyzed *in vitro* the effect of CM-272 treatment on cell proliferation in a large number of tumor murine cell lines. Unfortunately, as opposed to the results that we found in human derived tumor cell lines (Supplementary Table 9), CM-272 had a very limited inhibitory effect on the proliferation of CT26 and MC-38 (colon cancer cell), Hepa129 and Hep1_6 cells (hepatoma cells), 4T1 cells (breast cancer), B16OVA cells (melanoma), A20 cells (sarcoma), TC1 cells (lung cancer), 5TGM1 cells (Multiple Myeloma), EG7 cells (lymphoma) or C1498 cells (AML) (see figure below). Besides, this limited effect was obtained despite using very high concentrations of CM-272 (up to 10 μ M). In agreement with this lack of effect, we did not detect Calreticulin expression in murine cell lines after their culture in the presence of CM-272 (see figure below where production of DAMPS after doxorubicine but not after CM-272 is demonstrated in murine cell lines). These results might suggest that G9A/DNMT are playing different roles in mice and in human tumor cells. Indeed, it has been recently described that DNMT1 silencing, while resulting in rapid cell death in human embryonic stem cells, did not affect to the same extent the viability of murine cells, which remained viable after inhibition of DNMT1 (Liao et al Nat Genet, 2015; 47: 469-78). Similarly, opposite effects have been very recently described for G9a in humans and mice regarding its ability to regulate p53 activity (Rada et al, Oncogene, 2016 Jul 25. doi: 10.1038/onc.2016.258). In this study, the authors demonstrated that expression of human G9a augments the p53-dependent cell death, while opposite conclusions were drawn for murine cells. These differences between murine and human tumor cells in the biological consequences of inhibiting G9a or DNMT1, together with the resistance

of mouse-derived tumor cell lines to CM-272 treatment, further complicates the assessment of the antitumoral activity of this compound in immunocompetent murine models. Recent preliminary information in our lab, suggests that there might be additional murine immunocompetent models amenable to address this question so we are pursuing these models. Unfortunately, it will take several months before we can have a more definitive answer.

Effect of CM-272 treatment in murine cell lines. (A) GI50 values of CM-272 for the CT26 (colon cancer), A20 (sarcoma), 5TGM1 (Multiple Myeloma), Hepa129 and Hep1_6 (hepatoma cells), 4T1 cells (breast cancer), B16OVA cells (melanoma), MC-38 (colon cancer), EG7 (lymphoma) and TC1 (lung cancer) murine tumor cell lines. (B) Calreticulin exposure determined by FACS analysis of CT26 (colon cancer), A20 (reticulum cell sarcoma), 5TGM1 (myeloma) and C1498 (acute myeloid leukemia) murine tumor cell lines after 48h of treatment with Doxorubicine and CM-272.

I have no concerns regarding referencing, quality of work otherwise. Multiple *in vivo* replicates from each tumor model are absolutely required for robust data and reproducibility.

Following the Reviewer instruction, we have included in the revised version of the manuscript that the *in vivo* results were obtained in two biological replicates per cell line (Page 12) and we have included the results obtained in the second replicate per each hematological malignancy model as a Supplementary Figure S10. The robust reproducibility of *in vivo* results obtained with the treatment of CM-272 in two independent replicates per cell line and in three different hematological malignancy models (ALL, AML and DLBCL), demonstrates that CM-272 exerts a potent anti-tumor

activity *in vivo* against different types of hematological malignancies by inhibiting the methyltransferase activity of both G9a and DNMTs.

Minor point - table S2 has an error in the uM text.

We apologize for this formatting error in table S2. We have corrected this error in the Supplementary Tables section of the revised version of the manuscript.

Table S2: LC₅₀ values on cytotoxicity assays

Cell Type	CM-272	CM-579
	LC ₅₀ (μM)	LC ₅₀ (μM)
THLE-2	1.78	1.30
PBMCs	1.90	7.39

Reviewer #2 (Remarks to the Author):

The study investigates the dual inhibition of the methyltransferases G9a and DNMT1, primarily using novel bispecific inhibitors. The authors show a wealth of in vitro and cell-based data, providing evidence for the on-target effects on these inhibitors on AML cell viability.

Although inhibitors of the individual enzymes have been described, this is the first report of a bi-specific inhibitor, which has enabled an in-depth analysis of the combined effects of inhibiting both enzymes.

Overall the paper is very clearly written and the data is presented well. I applaud the level of detail given on the methods used in the study - this made it very easy to check that the methods used were appropriate and will help other groups who may wish to reproduce some of the experiments.

We are sincerely grateful to the reviewer for his constructive comments. Undoubtedly, these comments helped us to improve the quality of our manuscript and reach out to the target audience more adequately and effectively.

There is scope for some improvement:

1. The methyltransferase inhibitors identified in the study are clearly related to known inhibitors of G9a such as UNCO638. However, in contrast to the previous compounds, CM-272 also inhibits DNMT1. Based on the figures presented in the manuscript, it is difficult to see whether the binding sites for this compound on G9a/DNMT1 are related, and so the basis for dual specificity is unclear. This is because the views shown in Figure 1C and D are very different (as indicated by the different orientations of SAH). The authors should clarify whether the binding sites are structurally related, and include a superposition of the models. I also recommend that the authors include a description of the structural features that underpin the selectivity of the compound compared to related enzymes.

We acknowledge that binding modes described in Figure 1 are not clear enough to provide a comparison of the substrate binding site of DNMT and G9a. In fact, both targets are structurally divergent at the substrate binding site and also at the cofactor (SAM) binding site. Then, as discussed below, we provide additional information and figures to clarify this point.

Concerning G9a, CM-272 binds presumably at the SET domain of G9a, sandwiched by helix α F-strands β 10– β 11 and helix α Z (the last helix before the post-SET zinc binding domain). This helix α Z is unique to G9a and GLP (Chang Y. Structural basis for G9a-like protein lysine methyltransferase inhibition by BIX-01294. *Nat Struct Mol Biol.* 2009, 16(3):312-7). As already highlighted by these authors, other HKMTs have insertions in this region, which might be a cause for the selectivity of CM-272 against G9a/GLP over other HKMTs in Figure 1, as indicated above by the reviewer.

Concerning DNMT1, our docking studies and experimental competition assays (Figure 1; Results section page 6) suggest that CM-272 binds at the DNA binding site. Detailed analysis of docking studies proposes that CM-272 binds in the cavity conformed by seven-stranded β -sheet and three α -helices, more particularly, close to the α helix with the catalytic cysteine. Moreover, the role of the extremely flexible autoinhibitory linker cannot be fully established with the currently available crystallographic information. As expected, our molecules are not selective for any DNMT isoform (Table S4B).

Taking into account the different folding of G9a and DNMT1 at the substrate binding site, structural alignment of the substrate binding site of both targets is not straightforward. We have superposed both targets by superimposing the different ligands (SAH or CM-272) to provide a clear comparison

of the sites (Figure S3, included below). As shown below, both sites share little structural similarity; however, according to docking studies, our molecules fit and interact with the receptor in both cases (G9a and DNMT1) to achieve acceptable inhibitory capacities (confirmed by their IC_{50} values and suggested by competition studies). The proposed explicit interactions, CM-272 fitting to each binding site, are described in Figure S2 (included below).

Figure S3. Comparison of binding modes of CM-272 into G9a (orange) and DNMT1 (yellow). Given that both proteins are structurally divergent, docked ligand-protein complexes were superposed by the chemical structure of SAH (A, upper figures) and CM-272 (B, lower figures) for the purpose of comparison. SAH and CM-272 are shown in blue and pink sticks, respectively.

Figure S2. CM-272 interaction with G9a and DNMT1. (A-B) According to docking studies, proposed binding modes lead to these key ligand-receptor interactions suggesting the chemical functionalities that should be borne by designed molecules to cover these critical pharmacophoric features. These 2D interaction plots were generated using the MOE program (Chemical Computing Group, <http://www.chemcomp.com/>), where (A) represents interaction between CM-272 & G9a and (B) CM-272 & DNMT1.

That being said, we will like to notice that UNC0638 was taken as a reference molecule for modelling strategy of dual G9a-DNMT1 inhibitors following the observation that UNC0638 (and not BIX-01294) displayed in fact high micromolar activity against DNMT1 (Table S3 and <http://thesgc.org/chemical-probes/UNC0638>). The crystal structure of UNC0638 in complex with G9a (3RJW.pdb) and of BIX-01294 with GLP (3FPD.pdb) and, as noted by the reviewer, the similarity between UNC0638 and CM-272/CM-579 (and posterior competition studies) are all supportive of the predicted binding mode for CM-272/CM-579 at the histone binding site of G9a. For DNMT1, predicted binding mode in Figure 1 is a proposal of the plausible binding mode that better explains internal SAR (e.g. activity of UNC0638 vs BIX-01294 against DNMT1 – Table S3, to be reported in due course for our internal chemical series) and results from competition studies.

In the revised version of the manuscript the following changes have been made: a modified version of Figure 1 and a new supplementary Figure S3. Changes in the manuscript have been included in the results section (page 6).

2. The study is based on the hypothesis that dual inhibition of G9a and DNMT1 is a potent combination in killing cancer cells. Early in the text, a combination study is reported using inhibitors of DNMT1 (Decitabine) and G9a (A-366) on OCI-AML-2 cell line. This shows synergy, but it is difficult to compare these results with later experiments (using the bispecific inhibitors) because the data that underpins the CI calculations is not shown. I request that this data is included in a revised manuscript.

A central hypothesis of our study resides in the synergistic effect of dual inhibition of G9a and DNMT1 so indeed our results should support the hypothesis and in that sense we apologize for not providing all the information in a clearer way. In the revised version of our manuscript (new Figure S1, see below) we have now included the cell proliferation data (two replicates) that underpins the CI calculations obtained in our experiments with DNMT1 (Decitabine) and G9a (A-366) inhibitors (Figure S1A-B). As can be observed in Figures S1A and B, the combination of DNMT1 (Decitabine) and G9a (A-366) inhibitors induced a greater synergistic inhibition of cell proliferation than any single compound at any tested concentrations.

Figure S1. Combination study of G9a (A-366) and DNMTs (Decitabine) inhibitors. **A)** Cell proliferation data of A-366 and Decitabine alone or in combination at 12.5, 25, 50 and 100 μM of each of the drugs in OCI-AML-2 AML cell line. Combination of DNMT1 (Decitabine) and G9a (A-366) inhibitors induced a greater inhibition of cell proliferation than any single compound at any tested concentrations. Results of two biological replicates are shown. **B)** At any tested concentrations the CI is always lower than 1; according to ranges of CI defined by Chou (Chou TC, Pharmacol Rev, 2006): Black: moderate synergism (CI between 0,7 and 0,85); Blue: synergism (CI between 0,3 and 0,7); Salmon: strong synergism (CI between 0,1 and 0,3). Combination study set-up was based on GI_{50} values for assayed molecules, A-366 and Decitabine, vs tested cell line, their corresponding GI_{50} s are $> 50\mu\text{M}$ in both cases. Left: synergism level of four different combinations; Right: synergism level of all of the combination tested. A representative example of three different experiments is shown.

To further strength our hypothesis, we performed new studies of inhibition of DNMT1 and G9a using siRNAs. Specifically, we used two different siRNAs for each gene in three cell lines (CEMO-1, MV4-11 and OCI-AML-2). Both G9a and DNMT1 interference separately led to a cell proliferation inhibition in all cell lines tested. However, the combination of G9a and DNMT1 siRNAs induced a significantly greater inhibition of cell proliferation in comparison with any siRNA separately (see figure below). This experiment with siRNAs supports with a different strategy the synergistic effect on inhibition of cell proliferation by the dual inhibition of G9a and DNMT1, representing a novel approach for cancer therapeutics.

Figure S1C. Cell proliferation analyzes using specific siRNAs against G9a (1 and 2) and DNMT1 (A and B) alone or in combination in CEMO-1, OCI-AML-2 and MV4-11 and cell lines. *= $p < 0.05$; **= $p < 0.01$; ***= $p < 0.001$ and n.s.=no significant.

We have included these studies in the Results section (page 4), Supplementary Materials section (Page 13), and we have included these figures as Figure S1 of revised version of the manuscript.

3. There remains a lingering doubt in my mind whether the transcriptional changes and cell death induced by CM272 is due to the bi-specific inhibition of G9a/DNMT1 or whether the inhibition of other enzymes (such as PRMT1,6 and/or GLP) might also contribute. This possibility is dismissed too easily by the authors (final sentence of the Results). To address this, I suggest the authors carry out studies using genetic knockdown (e.g. RNAi) in combination with chemical inhibitors in a selected set of cell lines from their panel to determine whether other epigenetic targets might also contribute to immunogenic cell death.

This is indeed an interesting issue raised by the reviewer and is certainly something that we have discussed thoroughly and was introduced in the original version of the manuscript in the discussion section (Pages 13-15). Following the reviewer suggestions we have further analyzed the potential for inhibition of other epigenetic targets. In our biochemical studies of activity of CM-272 and CM-579 over a wide range of 37 epigenetic targets implicated in the regulation of epigenetic mechanism that we included in the initial submitted version of the manuscript, we observed that the percentage of inhibition of CM-272 and CM-579 at concentration of 10 μ M was 75 and 25% for GLP, 73 and 43% for PRMT1 and 82 and 40% for PRMT6, respectively (Supplementary Table S4A). Following these studies, the biochemical IC_{50} values of CM-272 and CM-579 against these targets were 2nM and >10000nM for GLP, 4000nM and >10000nM for PRMT1 and 3000nM and >10000nM for PRMT6, respectively using a chemiluminescence assay carried out at BPS (Supplementary Table S4B). Of note, using a radioligand binding assay (3 H-SAM, tritiated SAM), IC_{50} values of CM-272 and CM-579 against GLP are 7 nM and 67 nM, respectively. This new data has been conveniently indicated in Results section (Page 6) and Supplementary Table S4B of the revised version of the manuscript. Thus, with the exception of GLP, CM-272 and CM-579 exhibit a minimum difference of 1 log unit selectivity for G9a/DNMT over the PRMT targets (IC_{50} of CM-272 and CM-579 values of 8 and 16 against G9a and of 382 and 32 against DNMT1, respectively, Supplementary Table S3 of the initial submitted manuscript). Moreover, we would like to highlight that IC_{50} values against both PRMT targets are above the GI_{50} values for both compounds in cell lines of different hematological malignancy (CEMO-1. IC_{50} of CM-272: 218nM and IC_{50} of CM-579: 75nM; MV4-11. IC_{50} of CM-272: 269nM and IC_{50} of CM-579: 385nM; OCI-Ly10: IC_{50} of CM-272: 455nM and IC_{50} of CM-579: 31nM; Supplementary Table S3).

That being said, and following the suggestion of the reviewer, we have also carried out gene interference studies using specific siRNAs for *GLP*, *PRMT1* and *PRMT6* in 4 ALL cell lines (CEMO-1, PEER, SEM and LAL-CUN-2). As seen in the figure below, *PRMT1* and *PRMT6* gene expression inhibition had no significant effect on cell proliferation of ALL cells. Thereby, and considering that CM-272 and CM-579 should be administered in the micromolar range to achieve significant inhibition (50%) of both PRMT targets, we strongly believe that we can establish that the contribution of both targets to the observed anti-tumor effect of our compounds is negligible.

In a further step, and despite CM-272 and CM-579 not being inhibitors of other known methyltransferases (<10% at 10 μ M, Supplementary Table S4A), we examined the potential role of other known methyltransferases implicated in the methylation of the H3K9 histone mark such as SUV39H1, SUV39H2 and SETDB1. Therefore, we carried out gene expression inhibition studies using specific siRNAs for these genes in 4 ALL cell lines (CEMO-1, PEER, SEM and LAL-CUN-2). As shown in the figure below the interference of any of these genes did not induce a significant inhibition of cell proliferation (with the exception of PEER cell line). In contrast, *GLP* knockdown by siRNA in 4 ALL cell lines was associated with a decrease in cell proliferation in two of the four cell lines (see Figure below), but in this case we did not find any expression change of ISG genes that we observed with the CM-272 treatment (see Figure below). This results show that contribution of GLP to immunogenic cell death that we detect with CM-272 treatment is limited. Thus, considering these results and the low IC₅₀ values of CM-272 and CM-579 against GLP, we cannot fully discard the contribution of inhibiting GLP on the effect observed in leukemia cells lines. Therefore, we have conveniently modified this paragraph in Result (page 13) and Discussion section (page 13-15) of the revised version of the manuscript.

Cell proliferation time course in CEMO-1, PEER, SEM and LAL-CUN-2 ALL derived cell lines after nucleofection with siRNAs against PRMT1, PRMT6, SUV39H1, SUV39H2, SETDB1 and GLP. A representative example of three different experiments is shown.

Expression level analysis of ISGs in MV4-11 cell lines by Q-RT-PCR after 48h of CM-272 treatment or siRNA against GLP.

Reviewer #3 (Remarks to the Author):

The manuscript by Prosper, et al describes the discovery of reversible dual inhibitors of G9a and DNMTs and characterization of these inhibitors in cellular and in vivo models of hematological malignancies. There are a number of major issues noted below. In particular, the lead compounds CM-272 and CM-579 have not been convincingly characterized as true inhibitors of DNMTs. Therefore, it is questionable that the observed effects in cell-based and in vivo studies are truly due to pharmacological inhibition of G9a and DNMTs. In light of these major concerns, this manuscript is not suitable for publication in a high impact journal such as Nature Communications.

Major issues:

1. Characterization of CM-272 and CM-579 as inhibitors of DNMTs (DNMT1, DNMT3A and DNMT3B) is rather weak, far from convincing. These compounds showed inhibitory activities in only two biochemical assays (TR-FRET and tritiated SAM). Direct binding of these inhibitors to DNMT1, DNMT3A and DNMT3B should be established by biophysical methods such as ITC and SPR, but is missing. The claim that CM-272 and CM-579 bind to the DNA binding site of DNMT1 is questionable. The MOA studies showed that the compounds are competitive with a DNA substrate, but this result does not tell where the inhibitors bind. Docking studies are not sufficient to establish the binding mode. Either an X-ray cocrystal structure or an NMR structure is needed to convincingly demonstrate that these inhibitors bind to the DNA binding site of DNMTs. A crystal or NMR structure of DNMT1 or DNMT3A in complex with one of the lead compounds would also convincingly demonstrate that the compounds are true inhibitors of DNMTs.

We acknowledge the concerns raised by the reviewer. From our understanding, there are two main points to discuss: i) whether our compounds are true inhibitors of DNMTs and ii) its predicted binding site and mode. We honestly believe that point i) is well-supported by the enzymatic assays based on three different technologies and carried out at different laboratories (two of them by two independent external laboratories: chemiluminescence-based at BPS Bioscience (<http://bpsbioscience.com/>) and radioligand binding assay at Reaction Biology Corp (<http://www.reactionbiology.com/>)), as reported in the manuscript. Moreover, IC₅₀ values of other well-established weak DNMT inhibitors (e.g SGI 1027) are reproduced by using our internal assay. On the other hand, and as suggested by the reviewer, determination of direct binding with biophysical methods is very desirable; in fact, we are currently working in collaboration with Structural Genomics Consortium (SGC, in Toronto) to achieve the corresponding structural information: CM-272 and/or CM-579 co-crystallized with G9a & DNMT1 (contact person is Dr Peter Brown, peterj.brown@utoronto.ca). This experimental work is on-going but will take its time; as the referee might be aware of, crystallization of DNMT structures is particularly difficult and was recently reported (apo, ligand-free, form; Science (2011) **331**: 1036-1040).

In addition, we have also utilized MicroScale Thermophoresis (MST) as an easy and precise biophysical method to quantify biomolecular interactions between CM-579 and DNMT1 (full length). This is an immobilization-free assay, close-to-native conditions; in fact, pretty similar to the biochemical assay (same buffer). DNMT1 was labeled with Labeling Kits provided by NanoTemper Technologies, where the fluorescent red dye NT-647 was coupled via NHS coupling, and its concentration was kept constant; about 20nM.

For performing this experiment, we have kept the concentration of NT-647 labeled DNMT1 constant, while the concentration of the non-labeled CM-579 (CM-579, batch 2) was varied between 10 μ M – 0.3 nM. The assay was performed in buffer containing Tween20 (0.05%) and DMSO (0.1%). After a short incubation the samples were loaded into MST NT.115 standard glass capillaries and the MST analysis was performed using the Monolith NT.115 instrument. A K_d of 27 nM was determined for

this interaction between DNMT1 and CM-579; thus, a direct binding of this inhibitor to DNMT1 was established.

DNMT1 vs CM-579_2:

Concentrations on the x-axis, corresponding to CM-579, are plotted in nM. This assay was performed by NanoTemper Technologies (<http://www.nanotemper-technologies.com/>).

Concerning point ii), we fully agree with referee concerns about the quality/accuracy of modeling techniques in predicting ligand-protein complexes. Of note, we do not state that the proposed binding mode (especially for DNMTs) is experimentally determined, but conveniently tune the tone by talking of “plausible binding modes suggested that CM-272 and CM-579”.

2. It is surprising that authors have not conducted cellular studies by knocking down both G9a and DNMTs via shRNA or siRNA, or introducing catalytically inactive G9a and DNMTs via CRISPR/Cas9. This genetic approach, which is complementary to the pharmacological approach, should be performed to support the conclusion of the manuscript.

We completely agree with the reviewer and thank him/her for the comment. Following his/her suggestion we have carried out interference studies of both G9a and DNMTs using siRNAs. Specifically, we have used two different interferences for each gene in three cell lines (CEMO-1, OCI-AML-2 and MV4-11). Both G9a and DNMT1 interference separately led to a cell proliferation inhibition in all cell lines tested. However, the combination of G9a and DNMT1 siRNAs induced a significantly much greater inhibition of cell proliferation in comparison with any siRNA separately (see figure S1C below).

The experiment with siRNAs is consistent with results found when pharmacologically inhibiting both targets, as included in the original submitted version of the manuscript. In summary, knocking down G9a and DNMTs by siRNAs or by a pharmacological approach combining the DNMT1 (Decitabine) and G9a (A-366) inhibitors, undoubtedly demonstrates that the simultaneous or dual inhibition of G9a and DNMT1 is a potent combination that inhibits cell proliferation in human cancer cells and represents a novel approach in cancer therapeutics.

Based on these results, we have added these new results in Results section (page 4), Supplementary Materials section (Page 13), and we have included these figures in the Supplementary Figures section as Figure S1C of revised version of the manuscript.

Figure S1C. Cell proliferation analyzes using specific siRNAs against G9a (1 and 2) and DNMT1 (A and B) alone or in combination in CEMO-1, OCI-AML-2 and MV4-11 and cell lines. *= $p < 0.05$; **= $p < 0.01$; ***= $p < 0.001$ and n.s.=no significant.

- The toxicity of the lead compounds is also a major concern. CM-272 has an LC₅₀ of 1.8 μM in THLE-2 (a non-tumoral hepatic cell line) and LC₅₀ of 1.9 μM in PBMCs. CM-579 displayed similar LC₅₀s. It appears that these inhibitors are generally toxic (Table S9). At best, they have a very small therapeutic window (~ 1 log unit), which is insufficient. The maximum tolerated dose for CM-272 and CM-579 in mice is 2.5 mg/kg and < 1 mg/kg, again suggesting that these inhibitors are toxic. CM-272 also inhibited multiple kinases (Table S16).

We fully agree with the reviewer, toxicity is an important concern and the therapeutic window (around 1 log unit) is insufficient for a pre-clinical candidate. However, we are at the initial stage of drug discovery process (de-risking it); in fact, our aim is the identification of a chemical probe to perform *in vitro* and *in vivo* validation of a novel mode of action (MoA): dual G9a & DNMT1 inhibition. Nevertheless and despite the modest therapeutic window, the *in vivo* results clearly demonstrate therapeutic efficacy in models of acute leukemia, even as a single treatment.

Our main goal was *in vivo* proof-of-concept of this novel strategy/MoA; then, after hit identification of dual inhibitors (first-in-class), we worked to achieve an adequate chemical probe to perform an *in vivo* validation. Thus, we synthesized several molecules (> 100 compounds, as described in patent WO2015192981, corresponding SAR/SPR studies will be published at its due time) and we identified CM-272, based on its primary activities as well as ADME/Tox profiling and pharmacokinetics, as lead molecule to perform *in vivo* test; in fact, due to the fact that CM-579 does not meet some key criteria (PK parameters, as described in the manuscript) it was not selected for *in vivo* testing. Therapeutic window for CM-272 (1 log unit aprox) is adequate in terms of efficacy and safety for a long term treatment (28 days) in animal models at 2.5 mg/Kg, as our simulation based on experimental PK data suggested (Figure S9), and *in vivo* tests confirmed (Figure 4, Figures S9-S11); thus, despite the modest therapeutic window, the *in vivo* results with CM-272 clearly demonstrate therapeutic efficacy in models of acute leukemia, even as a single treatment.

On the other hand, after testing CM-272 at 10 μ M vs 97 kinases, our lead molecule only showed a percentage control < 1% for 7 kinases (assay details at Fabian MA, et al Nat. Biotechnol. 23, 2005, 329-336); then, the subsequent detailed K_d determination against these 7 kinases showed that CM-272 only hits 1 kinase (ALK) with K_d<1 μ M (K_d is 820 nM). According to PK data, at selected dose (2.5 mg/Kg), C_{max} for CM-272 is 406 nM; then, our lead molecule will minimally interact with kinases *in vivo*.

In conclusion, our lead molecule CM-272 is efficacious and safe *in vivo* at selected dose (no toxicity issue) and led us to achieve our main aim: MoA validation. However, as raised by reviewer, we still have to optimize our lead compound (improve its therapeutic window as well as other properties) to identify a plausible pre-clinical candidate for first-in-human clinical trial. Thus, the corresponding lead optimization process is currently on-going in our lab.

4. PK/PD relationship needs to be established for the *in vivo* efficacy studies. Although H3K9me2 and 5mC levels were assessed at one time point (Fig. S4F), drug levels should be measured for the treated animals to correlate with the reduction of H3K9me2 and 5mC and with phenotypic effects.

Indeed, PK/PD relationship is a critical point for any drug discovery project. In this case, epigenetic marks (H3K9me2 and 5mC) should be utilized to monitor *in vivo* efficacy from a mechanistic point of view.

As we have already demonstrated *in vitro*, in ALL, AML and lymphoma derived cell lines, reported molecules reduce H3K9me2 levels after treatment in a dose response manner (monitored by Western-blot; e.g. Figure 2B-D and S5C) and induce DNA demethylation in the promoter region of genes such as *CDKN2A* and *POU4F2* (monitored by pyrosequencing; e.g. Figure S6).

Pharmacokinetic (PK) studies are performed and drug levels measured in the treated animals, as reported in Figure S11C and Table S13. In addition, following your suggestion, we have also measured drug levels 1 week and 3 weeks after the treatment started (Figure below); these experimental results confirm the simulated data we utilized to select this dose of 2.5 mg/kg *in-vivo* (Figure S9A).

Figure. In red experimental data, complete PK curve for day 1 as well as CM-272 concentration measured 1 hour and 6 hours after compound administration at days 7 and day 21 of treatment. In blue, experimental PK curve determined for day 1 and transferred to days 7 and 21; these blue curves perfectly fit the experimental data (in red) obtained for those two days.

However, pharmacodynamic (PD) monitoring of histone or DNA methylation is currently a challenge from two perspectives: technical (appropriate MAb to develop and implement the corresponding quantitative assessment) and mechanistic (this is not the classical enzymatic response with a “fast” read-out: inhibition-response). In fact, as reported very recently by Epizyme based on its clinical trial for the EZH2 inhibitor Tazemostat (American Society of Hematology Meeting on Lymphoma Biology, in Colorado Springs, June 2016: “Chromatin flow cytometry based assessment of H3K27me3 pharmacodynamics in blood from diffuse large B-cell lymphoma and follicular lymphoma patients following exposure to the EZH2 inhibitor tazemetostat reveals disparate response profiles in specific PMBC subpopulations”; Poster #38 by C. Plescia et al), H3K27me3 inhibition is detected at day 15 or 30 of exposure to Tazemostat; therefore, EZH2 inhibition does not lead to a functional response that can be immediately monitored *in vivo*. In addition, PBMC subpopulations demonstrate markedly divergent H3K27me3 PD profiles; thus, this is not an obvious task. Finally, we should highlight that a specific flow cytometry based assay was developed and implemented by Epizyme to perform this monitorization; technical development, together with an appropriate MAb, was required to achieve a fine-tuned quantitative measurement of histone methylation – one step beyond western- and dot-blot.

Then, as we may expect based on these precedents shown by Epizyme, all our efforts to monitor inhibition of H3K9me2 or 5mC at day 1, day 7 or day 21 did not lead to any conclusion. But, as we previously reported (Figure S9F, included below), we were able to monitor a clear PD response by western- and dot-blot after 28 days treatment in a surrogate tissue (liver). This is a qualitative but unequivocal PD response. Thus, considering that PD monitorization of histone methylation involves a long process (exposure to inhibitor molecule for several days/weeks), the technical limitation for quantitative monitorization and PBMCs heterogeneity in PD profiles, we cannot provide a classical PK/PD plot for reported case studies; but, these PD results after *in vivo* treatment (Figure below) and PK data (Figure above), together with *in vitro* functional responses, suggest a clear correlation between reduction of H3K9me2 and 5mC with observed phenotypic effects.

Figure S9F. Liver H3K9me2 and 5mC levels after *in vivo* CM-272 treatment in CEMO-1 ALL mouse model. H3 total was used as a loading control.

5. Dose-dependent *in vivo* efficacy should be established, but is missing. The overall survival benefit is modest in all 3 models.

This is a very interesting point that we did not consider in our initial analyses. Following the suggestions from the reviewer, we have carried out an *in vivo* experiment with different doses of CM-272 in the ALL CEMO-1 model. 10×10^6 CEMO-1 cells diluted in 100 μL of saline solution were injected in the tail vein of 6-8 week old BALB/cA- Rag2^{-/-} $\gamma\text{C}^{-/-}$ mice. Three days after injection, treatment with 1mg/kg i.v CM-272 for 28 days was started in 6 mice, followed by a 14 days washout period. We controlled the mice body weight, hematological parameters and we measured the CM-272 plasma concentration. Besides, we also analyzed the same parameters in other 3 mice treated with 2.5mg/kg of CM-272 as shown in the manuscript. As expected, and in agreement with our previous results (included in the initial submitted version of the manuscript), we did not observe differences in the body weight of the animals nor significant changes in hematological parameters (WBC, RBC, PLT, hemoglobin and hematocrit) between mice treated with either concentration of CM-272 and the control group (saline solution). Moreover, CM-272 plasma concentration was greater in the mice group treated with 2.5mg/kg of CM-272 than in the mice group treated with 1 mg/kg mice. In contrast with mice treated with 2.5 mg/kg, no increase in overall survival was observed in mice treated with 1 mg/kg. These results indicate a dose-related effect at least in this model.

These results have been included in the revised version of the manuscript (Figure S11) and in the results and discussion (page 11 and page 12).

Figure S11. CM-272 is dose-dependent. (A) Body weight in *in vivo* ALL model. Mean \pm SEM body weight in CEMO-1 mice treated with Control (Saline Solution), 1 or 2.5 mg/kg daily for 4 weeks followed by a day's washout period. (B) Hematological parameters in *in vivo* CEMO-1 ALL model treated with 1 or 2.5 mg/kg of CM-272. WBC: white blood cells; RBC: red blood cells; PLT: platelet count. Black: Control group with saline solution, Blue: 1 mg/Kg CM-272 treatment, Salmon: 2.5mg/Kg CM-272 treatment. (C) CM-272 plasma concentration in the CEMO-1 ALL treated with 1 mg/Kg or 2.5 mg/Kg of CM-272 and measured for four weeks plus 14 days of washout period. (D) Kaplan-Meier survival curves for evaluating the survival time of mice engrafted with ALL derived CEMO-1 cells treated with 1 mg/Kg of CM-272. Control: Saline Solution (diluent of CM-272).

Minor concerns:

1. CM-272 displayed high potency for EHMT1 (GLP) ($IC_{50} = 2$ nM). Most of cellular results in the manuscript were generated using CM-272 and all *in vivo* results were generated using CM-272. It is not clear why authors ignored CM-272's EHMT1 inhibitory activity. Potential contributions from inhibition of EHMT1 cannot be ruled out in most of the cellular and *in vivo* studies they performed.

This is indeed an interesting issue raised by the reviewer and is certainly something that we have discussed thoroughly and was introduced in the original version of the manuscript in the discussion section (Pages 13-15). Following the reviewer suggestions we have further analyzed the potential for inhibition of other epigenetic targets. In our biochemical studies of activity of CM-272 and CM-579 over a wide range of 37 epigenetic targets implicated in the regulation of epigenetic mechanism that we included in the initial submitted version of the manuscript, we observed that the percentage of inhibition of CM-272 and CM-579 at concentration of 10 μ M was 75 and 25% for GLP, 73 and 43% for PRMT1 and 82 and 40% for PRMT6, respectively (Supplementary Table S4A). Following these studies, the biochemical IC_{50} values of CM-272 and CM-579 against these targets were 2nM and >10000nM for GLP, 4000nM and >10000nM for PRMT1 and 3000nM and >10000nM for PRMT6, respectively using a chemiluminescence assay carried out at BPS (Supplementary Table S4B). Of note, using a radioligand binding assay (3 H-SAM, tritiated SAM), IC_{50} values of CM-272 and CM-579 against GLP are 7 nM and 67 nM, respectively. This new data has been conveniently indicated in Results section (Page 6) and Supplementary Table S4B of the revised version of the manuscript. Thus, with the exception of GLP, CM-272 and CM-579 exhibit a minimum difference of 1 log unit selectivity for G9a/DNMT over the PRMT targets (IC_{50} of CM-272 and CM-579 values of 8 and 16 against G9a and of 382 and 32 against DNMT1, respectively, Supplementary Table S3 of the initial submitted manuscript). Moreover, we would like to highlight that IC_{50} values against both PRMT targets are above the GI_{50} values for both compounds in cell lines of different hematological malignancy (CEMO-1. IC_{50} of CM-272: 218nM and IC_{50} of CM-579: 75nM; MV4-11. IC_{50} of CM-272: 269nM and IC_{50} of CM-579: 385nM; OCI-Ly10: IC_{50} of CM-272: 455nM and IC_{50} of CM-579: 31nM; Supplementary Table S3).

That being said, and following the suggestion of the reviewer, we have also carried out gene interference studies using specific siRNAs for *GLP*, *PRMT1* and *PRMT6* in 4 ALL cell lines (CEMO-1, PEER, SEM and LAL-CUN-2). As seen in the figure below, *PRMT1* and *PRMT6* gene expression inhibition had no significant effect on cell proliferation of ALL cells. Thereby, and considering that CM-272 and CM-579 should be administered in the micromolar range to achieve significant inhibition (50%) of both PRMT targets, we strongly believe that we can establish that the contribution of both targets to the observed anti-tumor effect of our compounds is negligible.

In a further step, and despite CM-272 and CM-579 not being inhibitors of other known methyltransferases (<10% at 10 μ M, Supplementary Table S4A), we examined the potential role of

other known methyltransferases implicated in the methylation of the H3K9 histone mark such as SUV39H1, SUV39H2 and SETDB1. Therefore, we carried out gene expression inhibition studies using specific siRNAs for these genes in 4 ALL cell lines (CEMO-1, PEER, SEM and LAL-CUN-2). As shown in the figure below the interference of any of these genes did not induce a significant inhibition of cell proliferation (with the exception of PEER cell line). In contrast, *GLP* knockdown by siRNA in 4 ALL cell lines was associated with a decrease in cell proliferation in two of the four cell lines (see Figure below), but in this case we did not find any expression change of ISG genes that we observed with the CM-272 treatment (see Figure below). This results show that contribution of *GLP* to immunogenic cell death that we detect with CM-272 treatment is limited. Thus, considering these results and the low IC_{50} values of CM-272 and CM-579 against *GLP*, we cannot fully discard the contribution of inhibiting *GLP* on the effect observed in leukemia cells lines. Therefore, we have conveniently modified this paragraph in Result (page 13) and Discussion section (page 13-15) of the revised version of the manuscript.

Cell proliferation time course in CEMO-1, PEER, SEM and LAL-CUN-2 ALL derived cell lines after nucleofection with siRNAs against PRMT1, PRMT6, SUV39H1, SUV39H2, SETDB1 and *GLP*. A representative example of three different experiments is shown.

Expression level analysis of ISGs in MV4-11 cell lines by Q-RT-PCR after 48h of CM-272 treatment or siRNA against GLP.

- CM-272 and CM-579 are structurally similar to previously reported inhibitors of G9a and EHMT1. They are likely true inhibitors of G9a and EHMT1. Nevertheless, confirmation using a biophysical method such as ITC or SPR should be performed. A crystal or NMR structure of G9a or EHMT1 in complex with CM-272 would convincingly demonstrate that the inhibitor occupies the substrate-binding site.

Indeed, this suggestion is a very good idea; in fact, as mentioned above, we are currently working in collaboration with SGC (Structural Genomics Consortium, in Toronto) to achieve the corresponding structural information: CM-272 and/or CM-579 co-crystallized with G9a & DNMT1.

You can contact Dr. Peter Brown (peterj.brown@utoronto.ca) to confirm this.

- The effect of CM-272 on other common histone marks (in addition to H3K9me2, H3K9me3 and H3K27me3) should be evaluated.

We totally agree with the reviewer and following his/her suggestion, beside H3K27me3, we have evaluated the effect of different concentrations of CM-272 on distinct histone marks by western blot: H3K36me3, H3K4me3, H3K79me3 and H3 acetylation. As shown in the figure below, CM-272 treatment from 5nM to 1000nM concentration does not modify significantly the levels of these histone modifications. These results suggest that CM-272 specifically inhibits the methyltransferase activity of G9a and has no effect on other methyltransferases nor modifies other histone marks.

These results have been included in the revised version of the manuscript (Figure S5D) in the Supplementary materials (page 13) and results (page 8).

Figure S5D: Effect of CM-272 in different histone modifications. H3K27me3, H3K36me3, H3K4me3, H3K79me3 and H3 acetylation levels after 48 hours of treatment with different doses of CM-272 in CEMO-1 cells. H3 total was used as loading control.

Reviewers' comments:

Reviewer #1 (Remarks to the Author):

The revised manuscript has addressed the number of issues that I had brought up in the initial review. Importantly, additional experimental details and replicates have now been added that provide confidence in the voracity of the results.

I have no further concerns with the manuscript and would recommend acceptance for publication.

Reviewer #2 (Remarks to the Author):

The authors have made a detailed and clear response to my previous comments and I am fully satisfied with the revised manuscript.

Reviewer #3 (Remarks to the Author):

I thank the authors for their efforts on addressing my concerns. Some of the issues I raised have been successfully addressed. For example, knockdown of both G9a and DNMT1 via siRNAs in several cell lines, the effect of CM-272 on other common histone marks, and potential contributions from inhibition of EHMT1 to the observed cellular and in vivo effects. The authors also attempted to address the PK/PD relationship issue I raised and explained the technical difficulties in assessing PD markers (e.g., H3K9me2, 5-mC) following in vivo treatment. I will not request additional experiments regarding this issue. The revised manuscript is significantly improved.

However, two key issues remain (see below). Based on these major concerns, I cannot support this manuscript in its current form to be accepted by Nature Communications.

Major issues:

1. The issue of demonstrating CM-272 and CM-579 as true inhibitors of DNMTs (DNMT1, DNMT3A and DNMT3B) was not addressed satisfactorily. There is no question that these compounds were active in DNMT biochemical assays. However, being active in biochemical assays does not mean that these compounds are true DNMT inhibitors. False positives are very common in DNMT biochemical assays. The authors used SGI-1027 as a positive control in their biochemical assay. In fact, SGI-1027 has NOT been convincingly demonstrated as a true inhibitor of DNMTs even though it was active in DNMT biochemical assays. The authors have generated binding data using MST (I do not understand why the authors did not include the MST data in the revised manuscript). This is a good step, but is insufficient. To convincingly demonstrate that CM-579 directly binds to DNMT1, a gold standard biophysical method (ITC or SPR) must be used. Alternatively, either a crystal or NMR structure of DNMT1 or DNMT3A in complex with CM-272 or CM-579 is needed to address this concern.

2. The toxicity of the lead compounds remains a major concern. CM-272 has an LC50 of 1.8 μ M in THLE-2 (a non-tumoral hepatic cell line) and LC50 of 1.9 μ M in PBMCs. CM-579 displayed similar LC50s. It appears that these inhibitors are generally toxic (Table S9). It is questionable that the observed effects in cell-based and in vivo assays are due to pharmacological inhibition of only G9a and DNMTs.

Response to the Reviewers' comments (manuscript NCOMMS-16-14716 by E. San José-Enériz *et al.*)

We thank the reviewers for their thorough revision and suggestions on our revised manuscript as well as for their positive feedback. We have added new information and new experiments to address the third reviewer's comments which we sincerely believe contribute to improve the quality of the manuscript.

Below you can find the detailed answers to all the issues raised by the third reviewer.

Reviewers' comments:

Reviewer #1 (Remarks to the Author):

The revised manuscript has addressed the number of issues that I had brought up in the initial review. Importantly, additional experimental details and replicates have now been added that provide confidence in the veracity of the results.

I have no further concerns with the manuscript and would recommend acceptance for publication.

We sincerely appreciate the reviewer's general comment.

Reviewer #2 (Remarks to the Author):

The authors have made a detailed and clear response to my previous comments and I am fully satisfied with the revised manuscript.

We sincerely appreciate the reviewer's general comment.

Reviewer #3 (Remarks to the Author):

I thank the authors for their efforts on addressing my concerns. Some of the issues I raised have been successfully addressed. For example, knockdown of both G9a and DNMT1 via siRNAs in several cell lines, the effect of CM-272 on other common histone marks, and potential contributions from inhibition of EHMT1 to the observed cellular and in vivo effects. The authors also attempted to address the PK/PD relationship issue I raised and explained the technical difficulties in assessing PD markers (e.g., H3K9me2, 5-mC) following in vivo treatment. I will not request additional experiments regarding this issue. The revised manuscript is significantly improved.

However, two key issues remain (see below). Based on these major concerns, I cannot support this manuscript in its current form to be accepted by Nature Communications.

We are sincerely grateful for the positive comments regarding our revised version of the manuscript, as well as for his/her constructive suggestions.

Below, we provide a detailed response to his/her main concerns.

Major issues:

1. The issue of demonstrating CM-272 and CM-579 as true inhibitors of DNMTs (DNMT1, DNMT3A and DNMT3B) was not addressed satisfactorily. There is no question that these compounds were active in DNMT biochemical assays. However, being active in biochemical assays does not mean that these compounds are true DNMT inhibitors. False positives are very common in DNMT biochemical assays. The authors used SGI-1027 as a positive control in their biochemical assay. In fact, SGI-1027 has NOT been convincingly demonstrated as a true inhibitor of DNMTs even though it was active in DNMT biochemical assays. The authors have generated binding data using MST (I do not understand why the authors did not include the MST data in the revised manuscript). This is a good step, but is insufficient. To convincingly demonstrate that CM-579 directly binds to DNMT1, a gold standard biophysical method (ITC or SPR) must be used. Alternatively, either a crystal or NMR structure of DNMT1 or DNMT3A in complex with CM-272 or CM-579 is needed to address this concern.

We fully agree with the reviewer that demonstrating that these molecules to be a DNMTs inhibitors is a critical issue in our manuscript. To demonstrate that our compound inhibit the methyltransferase activity of DNMTs, in addition to the three orthogonal biochemical assays that we showed in our previous version of the manuscript (Result section: Pages 5 and 6), we also monitored the changes in DNA methylation at cellular level after treatment of different cells with our dual inhibitors. These analyses revealed that CM-272 inhibits the DNMTs activity showing a clear **DNA demethylation** at global level and also in promoters of specific genes after treatment with CM-272:

i.- Figure 2B and C. Dot blot of 5-methylcytosine shows impact of CM-272 on 5mC levels *in vitro*.

Figure 2: CM-272 inhibits cell proliferation and induces apoptosis in AML, ALL and DLBCL cell lines. (B) H3K9me2 and 5mC levels after CM-272 treatment with different doses for 48 hours in CEMO-1 cell line. H3 total was used as loading control. (C) H3K9me2 and 5mC levels after CM-272 treatment with different doses for 48 hours in MV4-11 cell line. H3 total was used as loading control.

ii.- Figure S7. DNA methylation analysis by pyrosequencing in the promoter region of *CDKN2A* and *POU4F2* genes, in AML derived cell lines (MV4-11 and OCI-AML-2), clearly shows that CM-272 induces DNA demethylation.

Figure S6. DNA methylation analysis. (A-B) CM-272 induces demethylation in promoter region of *CDKN2A* and *POU4F2*. DNA methylation analysis by pyrosequencing of the promoter region of *CDKN2A* (A) and *POU4F2* (B) in AML derived cell lines MV4-11 and OCI-AML-2 and before and after treatment with CM-272. Data shown are a representative experiment of three independent experiments.

iii.- Figure S10F. Dot blot of 5-methylcytosine shows impact of CM-272 on 5mC levels *in vivo*.

Figure S10. (F) Liver H3K9me2 and 5mC levels after *in vivo* CM-272 treatment in CEMO-1 ALL mouse model. H3 total was used as a loading control.

However, these results may still not be considered enough and in that sense we have added additional studies in the new revised version of the manuscript. As very recently reported in Nature Drug Discovery Reviews (Oct 2016, 15, 679 – 698), Microscale Thermophoresis (MST) is part of a “selection of established biophysical methods for analysis of protein-ligand interactions”; in fact, “a binding experiment using SPR, NMR, NC-MS or **MST** can confirm that hits bind directly to the protein

target of interest". Therefore, we utilized this well-established biophysical method (MST) to confirm that our molecules bind DNMT1.

Following reviewer suggestion, binding data using MST is now included in the revised version of the manuscript (Page 7). We have repeated this experiment in triplicate and K_d (CM-579 vs DNMT1) is slightly better than initially determined: 1.5 nM (Figure S3). We have included the experimental procedure details in the revised version of the manuscript (Page 21) and revised version of Supplementary Materials section (Pages 11-12).

Figure S3. Concentrations on the x-axis, corresponding to CM-579, are plotted in nM. This plot represents data from three independent measurements ($n=3$). A K_d of 1.5 nM was determined for this interaction between DNMT1 and CM-579.

Finally, as we already mentioned in the previous response, we are currently working in collaboration with Structural Genomics Consortium (SGC, in Toronto) to achieve the corresponding structural information for CM-579 co-crystallized with DNMT1 (contact person is Dr Peter Brown, peterj.brown@utoronto.ca). This experimental work is on-going but will take its time; as the referee might be aware of, crystallization of DNMT structures is particularly difficult and was recently reported (apo, ligand-free, form; Science (2011) **331**: 1036-1040).

2. The toxicity of the lead compounds remains a major concern. CM-272 has an LC50 of 1.8 μ M in THLE-2 (a non-tumoral hepatic cell line) and LC50 of 1.9 μ M in PBMCs. CM-579 displayed similar LC50s. It appears that these inhibitors are generally toxic (Table S9). It is questionable that the observed effects in cell-based and in vivo assays are due to pharmacological inhibition of only G9a and DNMTs.

Indeed, poly-pharmacology is always an important challenge in drug discovery and might also play a role in this case; thus, comprehensive target identification is key for rationalizing a molecule's therapeutic and adverse effect.

Then, to identify potential "off-target" effects, we tested CM-272 and CM-579 *versus* 36 additional epigenetic targets; and CM-272, chemical probe for *in-vivo* PoC, was also tested against a selected panel of 97 kinases distributed through the kinome. In total, CM-272 was tested against >130 potential off-targets that mechanistically might be involved in anti-tumoral response or cytotoxic effects.

From our point of view, screening >130 additional targets may be enough to achieve our goal: *in-vivo* PoC for a first-in-class chemical probe (dual G9a & DNMT inhibitor). Indeed, if this molecule were the clinical candidate (this is not the case) an exhaustive off-target selectivity profiling would be required to progress to first-in-human.

Just as additional information a very recent article in Nature Chemical Biology (November 2016, 12, 908 – 910) reports a previously unknown off-target for Panobinostat (HDAC inhibitor, FDA approved drug - 2015); it is a potent phenylalanine hydroxylase inhibitor. Before this molecule progressed to clinical trials, a large off-target selectivity profiling was performed to Panobinostat; however, this additional target (phenylalanine hydroxylase) had not been identified. Thus, we believe the reviewer maybe right and additional targets might be also involved in CM-272 activity but, according to the goal of the project (*in-vivo* PoC), we sincerely believe we have profiled a very significant and reasonable number of additional targets; this will not ensure that we will identify additional targets (as happened to Panobinostat).

Reviewers' comments:

Reviewer #3 (Remarks to the Author):

I thank the authors for including the MST data in the revised manuscript, which make the manuscript improved. However, I am still concerned about the lack of ITC or SPR data. While I agree with the authors that obtaining a cocrystal structure of DNMT1 in complex with CM-579 is not an easy task and takes time, ITC or SPR experiments are straightforward and do not require a lot of efforts. Why can't the authors do these experiments and include the results in the manuscript? The data will address my remaining concern. The argument that CM-272 reduced DNA methylation in cellular assays misses the point. The effect of the compound on DNA methylation in cells could be indirect (i.e., not due to direct inhibition of DNMTs).

Response to the Reviewers' comments (Manuscript NCOMMS-16-14716 by E. San José-Enériz *et al.*)

We thank the Editor for the suggestion and positive feedback. We have added new information to address the third reviewer's comment.

Below you can find a detailed answer to issues raised by the third reviewer.

Reviewers' comments:

Reviewer #3 (Remarks to the Author):

I thank the authors for including the MST data in the revised manuscript, which make the manuscript improved. However, I am still concerned about the lack of ITC or SPR data. While I agree with the authors that obtaining a co-crystal structure of DNMT1 in complex with CM-579 is not an easy task and takes time, ITC or SPR experiments are straightforward and do not require a lot of efforts. Why can't the authors do these experiments and include the results in the manuscript? The data will address my remaining concern. The argument that CM-272 reduced DNA methylation in cellular assays misses the point. The effect of the compound on DNA methylation in cells could be indirect (i.e., not due to direct inhibition of DNMTs).

We completely agree with the reviewer that direct interaction between CM-579 and DNMT1 needs to be demonstrated using established biophysical techniques such as SPR, ITC or MST (Renaud, et al. *Nat Rev Drug Discov* 15:679-98, 2016). However, the technical key requirements of these 3 approaches are very different. For example, a) ITC needs much larger amounts than either SPR or MST, and more so if the enthalpic contribution to binding is small, which is unpredictable; b) SPR requires immobilization of the target protein; and c) MST may involve protein labeling (as it is the case here).

We did not attempt ITC experiments because the excessive amount of protein needed (the minimal concentration of protein in the cell of the calorimeter required for conducting a titration with high enough signal to noise ratio is typically 10 μ M, in a cell volume of 300 μ L, see Linkuviene et al, *Anal Biochem* 515:61-64 (2016)). In addition, if the interaction is entropically driven the enthalpic contribution can be very small and the signal be insufficient to follow the binding titration (in this scenario, higher amounts will be required). An even more unfavorable scenario would appear if the compounds display low- or sub-nanomolar affinity (as it is the case for our compounds, according to MST data), that require ITC experiments in competition mode with a previously characterized lower affinity ligand.

The requirement to produce DNMT1 protein in large quantity is indeed important for us for additional reasons. We are currently leading a European project to screen a large library of compounds *versus* DNMT1 ("European Lead Factory" financially supported by EU (H2020) and EFPIA and DNMT1 expression is a key objective. The Biochemistry department from Oxford University (our contact person is maciej.kliszczak@sgc.ox.ac.uk) has been working for more than one year to achieve an adequate expression level and this has not been possible yet (we are still investing a huge effort on it). We have been able to purchase small amounts of DNMT1 to perform biochemical activity assays, demanding a small quantity of enzyme; but, despite all our efforts, enough protein to perform ITC experiments has not been possible to obtain. This is the reason that has precluded us from doing ITC.

We have indeed tried to measure the affinity of the reported compounds for DNMT1 using SPR. For that, **we attempted SPR experiments** using the two different DNMT1 protein constructs we can purchase: A full length DNMT1 with an N-terminal GST fusion, and a shorter construct containing the DNMT1 catalytic domain (501-1632) with N-terminal polyhistidine and Flag tags.

First, we immobilized GST-tagged full length DNMT1 to a chip containing anti-GST antibody, but the amount immobilized was only 8% of the GST Ab chip capacity, and insufficient to detect binding of the ca. 500 Da compounds, presumably due to steric impediments resulting from the large size (dimer of 209 kDa monomers) of the GST-DNMT1 fusion protein. The theoretical SPR signal expected for the compounds binding with a 1:1 stoichiometry to the amount of protein immobilized (2300 RU) is only 5 response units (RU), with the precision of our BiacoreX100 SPR instrument typically being 1-2 RU. The compounds tested (CM272 and CM579) showed the same binding to the DNMT1 and reference channels at 5 μ M, **indicating unspecific binding to the chip surface/Ab/GST**, and very likely masking their possible direct binding to DNMT1. Compound injections at 200 μ M concentration resulted in a similar behavior, but with higher responses in both channels, and a differential increase in the DNMT1 channel that indicates non-reversible binding with a high stoichiometry (responses of 81 and 65 RU, respectively; with 5 RU expected for a 1:1 binding), likely resulting from **aggregation of the compounds at 200 μ M concentration in the DNMT1 channel**.

The second SPR strategy used the polyhistidine tag of the N-terminal DNMT1 catalytic domain (501-1632), DNMT1 truncated form, to retain it in an NTA SPR chip, after which it was covalently attached through lysine residues, and we were able to immobilize 12362 RU, that should give a response of ca. 48 RU for 500 Da compounds (for 1:1 binding stoichiometry). To assess the activity of the immobilized DNMT1 we titrated its cofactor SAM (507 Da) from 98 nM to 100 μ M and obtained a K_D =15-20 μ M and an stoichiometry close to 1:1, in agreement with the literature values (figures below). However, repetition of the experiments resulted in a decrease of the stoichiometry (from 1 to 0.8 in 24 h), indicative of DNMT1 inactivation for SAM binding with time. Compounds CM-579 and CM-272 were titrated after the first SAM titration on the same freshly prepared chip at concentrations also ranging from 98 nM to 100 μ M. For CM-579 very low differential responses are obtained at sub-micromolar concentrations, and **the compound also binds to the reference channel, masking the detection of binding to DNMT1**. For CM-272 a similar behavior was observed. In conclusion, in this case, the SPR approach was not an appropriate biophysical strategy to determine interaction affinities between DNMT1 and the assayed molecules because the compounds bind the reference channel and/or also probably because immobilization of the DNMT1 through lysine residues may alter the compound (but not SAM) binding site in the enzyme.

For the reasons described above we have relied on MST results that in our opinion conclusively demonstrate the direct physical interaction between CM-579 and DNMT1 with low nanomolar affinity. MST has been recently demonstrated to be a well-established biophysical method as reliable as either SPR, NMR or NC-MS (according to the very recent review by Renaud, et al. (*Nat Rev Drug Discov* 15:679-98, 2016) describing that “a binding experiment using SPR, NMR, NC-MS or **MST** can confirm that hits bind directly to the protein target of interest”) and based on this we tried the MST approach; in this case, we did not experience any technical issue and binding affinity (K_D) was properly measured. Therefore, we trusted on this well-established biophysical method (MST) to confirm that CM-579 binds DNMT1.

Figure. SPR sensorgrams (left) of SAM flown at 98 nM to 100 μM concentrations over immobilised DNMT1 (501-1632), and corresponding fit of the steady state responses at each concentration to a binding isotherm with equivalent sites (right). Best fit values are indicated.

REVIEWERS' COMMENTS:

Reviewer #3 (Remarks to the Author):

I thank the authors for conducting SPR experiments. However, my lingering concern has not been addressed by the negative results of the SPR experiments and the lack of ITC experiments. The authors were able to generate SPR data for the cofactor SAM ($K_d = 15 - 20 \text{ } \mu\text{M}$), suggesting that the method is working. However, they were unsuccessful in demonstrating binding of CM-579 or CM-272 to DNMT1 using this method. It is possible that the immobilization of DNMT1 through lysine residues alters the compound, but not SAM, binding site of DNMT1, as pointed out by the authors. However, the negative results do not enhance the data package. They actually increased my concern. The authors previously stated that they are attempting to cocrystallize CM-579 or CM-272 with DNMT1. Co-crystallization experiments require a large amount of the enzyme, more than what ITC experiments require. The authors ought to have enough amounts of DNMT1 for ITC experiments. They should do ITC experiments and aim to generate positive results.